# SHERPA: Fine-tuning Segment Anything Models with Task-relevant Guidance

Jingcheng Xie [1 2]  Yinda Chen [1 2]  Xiaoyu Liu [1 2]  Haoyuan Shi [1 2]  Zhiwei Xiong [1 2]

## Abstract

Segment Anything Models (SAMs) often struggle with certain specialized tasks. A common approach is to fine-tune models with specific task labels, but this often leads to overfitting, introduces model bias and significantly degrades their generalization ability. To overcome these challenges, we propose SHERPA, a novel framework that leverages a smaller SAM to guide the fine-tuning of a larger SAM via task-relevant features. Specifically, we first leverage the Fisher Ratio Separation (FRS) module to separate high task-relevant features and preserve the ability of the large SAM to perform other general tasks. Then, the Guiding Feature Extraction (GFE) module is used to extract representative guiding features from the fine-tuned small SAMs. We leverage small SAMs tailored for specific tasks (including natural image segmentation, biomedical image segmentation, and video object segmentation) as guidance and then evaluate the SHERPA scheme to fine-tune larger SAM series models. Our experiments demonstrate that SHERPA enhances the retention of generalization ability across those diverse tasks, by up to 11.1%, and improves specific task performance by up to 2.2%. Code: https://github.com/xiejingcheng/SHERPA

## 1. Introduction

*"A Sherpa's strength lies in the wisdom to navigate, not in overpowering the mountain."*

Recently, SAMs have emerged as foundation models for image segmentation, achieving remarkable success owing to their strong generalization capabilities (Kirillov et al., 2023;

Ravi et al., 2024). However, they often struggle with certain specialized tasks (Wu et al., 2023; Ke et al., 2024), such as biomedical image segmentation and fine-grained object segmentation. A widely adopted solution is fine-tuning these SAMs using specific task labels, adapting them to specific tasks (Andreassen et al., 2021; Cao et al., 2024). Despite its effectiveness, this process faces certain challenges. Fine-tuning generally relies on a limited number of labeled data, which can lead to overfitting to specific task data, thereby introducing model bias and resulting in a degradation of the model's original generalization ability (Li et al., 2020a). This presents a dilemma: without fine-tuning, the model underperforms on specialized tasks; with fine-tuning, it tends to degrade generalization ability.

This issue can be attributed to the compression of generalization-relevant information (Wortsman et al., 2022). According to the information bottleneck theory (Tishby & Zaslavsky, 2015), during fine-tuning, the model enhances the information that is highly relevant to the specific fine-tuning task, while compressing the remaining information that is less relevant. However, this inevitably results in a degradation of the generalization ability acquired during pre-training, because the compressed information is only weakly relevant to the current fine-tuning task, but may still contain substantial information relevant to other generalization tasks (Cai et al., 2024; Cao et al., 2024). For example, when fine-tuning on biomedical image segmentation, the model compresses information about natural images learned during pretraining, leading to a degradation of generalization ability.

To mitigate the degradation of generalization performance during fine-tuning, we decouple the optimization of task-relevant and generalization-relevant information. Specifically, we isolate those features that contain generalization-relevant information and restrict updates primarily to the task-relevant features during fine-tuning. By confining the fine-tuning process to task-relevant features, this approach minimizes interference with generalization-relevant information, thereby better preserving the model's generalization ability.

Therefore, our first step is to separate the task-relevant features from the rest. However, this separation is challenging because these task-relevant features are often stochastically

[1]MoE Key Laboratory of Brain-inspired Intelligent Perception and Cognition, University of Science and Technology of China [2]Anhui Province Key Laboratory of Biomedical Imaging and Intelligent Processing, Institute of Artificial Intelligence, Hefei Comprehensive National Science Center. Correspondence to: Zhiwei Xiong <zwxiong@ustc.edu.cn>.

*Proceedings of the 43rd International Conference on Machine Learning*, Seoul, South Korea. PMLR 306, 2026. Copyright 2026 by the author(s).

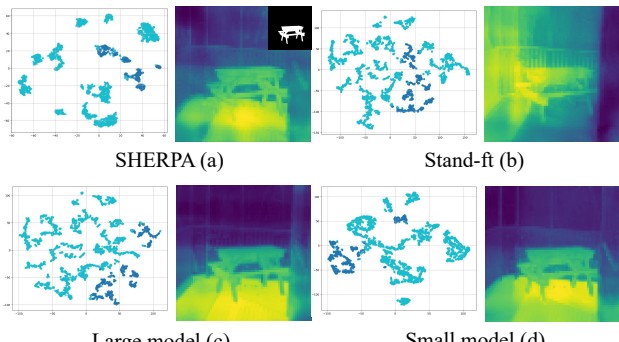

*Figure 1.* t-SNE visualizations compare feature distributions. Standard fine-tuning compresses generalization-relevant features, while the small model (d) captures more focused, task-relevant features with a higher Fisher ratio than the large model (c). The GFE module identifies and transfers these high-Fisher-ratio, task-relevant features from the small model to guide the fine-tuning of the large model, which helps recover generalization ability (as reflected by the improvement from b to a).

distributed across channels, lacking any discernible pattern. To address this, we introduce the Fisher ratio as a criterion to quantify task relevance. By maximizing this criterion in an optimized subspace, our Fisher Ratio Separation (FRS) module isolates a compact set of task-relevant feature components. Based on this, we propose a Fisher Ratio Separation (FRS) module that constructs a subspace containing these task-relevant features. While the FRS module enables us to effectively separate task-relevant features, further challenges remain in how to optimally guide the fine-tuning of task-relevant components in larger models. In particular, for the guidance to be most effective, it is desirable to obtain a more representative set of high Fisher ratio features.

Information bottleneck theory suggests that limited-capacity models are pressured to retain information that is most useful for the target task. In fine-tuning, restricted capacity can act as a stronger bottleneck, encouraging the model to concentrate its representation on a compact set of highly task-discriminative components. Consistent with this intuition, we observe that the small model tends to produce components with higher Fisher-ratio scores during fine-tuning, as illustrated in Figure 1, even though it may not achieve the best overall performance.

Motivated by this observation, we introduce the Guiding Feature Extraction (GFE) module. GFE extracts the high-Fisher, task-relevant components from a fine-tuned small SAM and uses them as guidance to steer the adaptation of a larger SAM. This guidance helps reduce interference with generalization-relevant information while improving task-specific adaptation.

Finally, we provide both theoretical analysis and empirical evidence to demonstrate the effectiveness of our SHERPA method. We fine-tune the larger SAM in 4 datasets (including natural image segmentation, biomedical image seg-

mentation, and video object segmentation) and evaluate generalization ability across 12 datasets, demonstrating that our SHERPA method effectively alleviates the degradation of generalization ability and improves task-relevant performance, with up to 11.1% improvement in generalization retention and 2.2% in task-specific performance across diverse tasks. We further extend SHERPA to other architectures, including SAM variants, MaskFormer, and DINO.

Our contributions are summarized as follows:

- We identify the loss of generalization ability in large SAM models during fine-tuning and propose to leverage task-relevant features from small SAMs to address this challenge.

- We design a two-stage framework to implement this idea: (1) a Fisher Ratio Separation (FRS) module that separates task-relevant features from other representations, and (2) a Guiding Features Extraction (GFE) module that extracts and transfers these features from the small SAM to guide the fine-tuning of the large model.

- We provide both theoretical analysis and empirical evidence to demonstrate the effectiveness of our SHERPA method, and further show its applicability to architectures beyond SAM.

## 2. Related Work

**Robust Fine-tuning.** Robustness is a critical challenge in deep learning, as fine-tuned models often lose generalization on unseen data (Torralba & Efros, 2011). In NLP, stable fine-tuning methods have been proposed to address representational collapse, though often at increased computational cost (Jiang et al., 2020; Zhu et al., 2020; Aghajanyan et al., 2021). In vision, regularization-based approaches are widely used to mitigate generalization loss, including sequential learning regularization (Kirkpatrick et al., 2017; Zenke et al., 2017), quadratic regularization (Li et al., 2018), and careful tuning of fine-tuning hyperparameters (Li et al., 2020a). Combining model weights has also been explored to improve generalization (Wortsman et al., 2022).

**Parameter-Efficient Visual Fine-tuning.** Parameter-efficient fine-tuning (PEFT) methods have gained popularity in vision. Visual prompt tuning (Jia et al., 2022) and Adapt-Former (Chen et al., 2022) adapt pre-trained models to new tasks with minimal additional parameters. Other approaches include low-rank adapters (Yin et al., 2023), sensitivity-aware parameter selection (He et al., 2023; Xie et al., 2026), gradient-based parameter selection (Zhang et al., 2024), and methods optimizing for memory and time efficiency (Yin et al., 2024). Recent work shows that tuning a subset of parameters can outperform full fine-tuning in visual recognition tasks (Yin et al., 2025).

**Large Segmentation Models.** Segment Anything Model (SAM) (Kirillov et al., 2023) is a foundational promptable segmentation model, with extensions to video (Ravi et al., 2024), fine-grained masks (Ke et al., 2024), and improvements in efficiency via distillation and quantization (Liu et al., 2024). SAM and its variants are widely used in applications such as medical imaging (Wu et al., 2023).

## 3. Method

### 3.1. THEORETICAL ANALYSIS OF SHERPA

Fine-tuning large SAM models requires identifying and preserving features that are most discriminative for the target task, while maintaining generalization. To this end, we use the Fisher ratio to measure class separability in feature space. Maximizing the Fisher ratio selects the most task-relevant features. We achieve this by projecting features into a subspace where the Fisher ratio is maximized, allowing us to focus fine-tuning on these components. The following provides our theoretical formulation. Detailed proofs are in Appendix A.

**Definition 3.1** (Feature Mapping Functions). Let $f_{\text{small}} : \mathcal{X} \to \mathbb{R}^d$ denote the feature mapping of a fine-tuned small SAM model, and $f_{\text{large}} : \mathcal{X} \to \mathbb{R}^d$ denote the feature mapping of a pretrained large SAM model. Let $g : \mathbb{R}^d \to \mathbb{R}$ be the final segmentation head.

**Definition 3.2** (High Fisher Ratio Subspace). Given $k < d$, and a sample set $\{(x_i, y_i)\}_{i=1}^m$, the high Fisher ratio subspace is defined as

$$W = \underset{W^\top W = I_k}{\arg\max} \, FR(W^\top f_{\text{small}}(x); y),$$

where $W \in \mathbb{R}^{d \times k}$, and $\bar{W}$ denotes its orthogonal complement in $\mathbb{R}^d$. Here, the function $FR(\cdot, \cdot)$ denotes the Fisher ratio, which is defined as the ratio of between-class variance to within-class variance for the features. A higher Fisher ratio implies stronger class separability, making it a suitable metric for identifying features that are most predictive and useful for the target task.

**Assumption 3.3** (Finite Second Moment). For any input $x \sim \mathcal{D}$, both the feature representation of the small model, $f_{\text{small}}(x)$, and the initial feature representation of the large model, $f_{\text{large}}^0(x)$, have bounded squared $\ell_2$ norm:

$$\mathbb{E}\left[\|f_{\text{small}}(x)\|_2^2\right] \le B^2, \qquad \mathbb{E}\left[\|f_{\text{large}}^0(x)\|_2^2\right] \le B^2.$$

This assumption guarantees bounded feature representations, a condition commonly met in practice by normalization layers (e.g., LayerNorm) and regularization techniques in SAM models.

**Assumption 3.4** (Capacity Control and Lipschitzness). Assume that the final segmentation head has model capacity

bounded by $C_w$. Furthermore, assume that the feature extractor is $L_f$-Lipschitz with respect to its parameters. This assumption controls model complexity and ensures smooth changes in features during parameter updates, which is consistent with the design of ViT and SAM architectures.

**Theorem 3.5** (Generalization Bound Comparison: SHERPA vs Standard Fine-tuning). *Let $m$ samples be used to estimate $W$, and $n$ samples for fine-tuning ($m + n = N$). Then the following inequality holds:*

$$R(\theta_{\text{SHERPA}}) \le R(\theta_{\text{FT}}) + \underbrace{\tilde{\Delta}_W}_{\text{subspace selection}} + \underbrace{\frac{C\,C_w^2 B^2}{\sqrt{n}}\left(\sqrt{k} - \sqrt{d}\right)}_{\text{capacity gain}}$$

*where $R(\cdot)$ denotes an upper bound on the generalization risk of the model, and $\tilde{\Delta}_W = O(B\sqrt{k \log d/m})$ quantifies the subspace estimation error. In particular, if $m \ge \frac{a^2\,n\,k\,\log(d/\delta)}{C^2\,C_w^4\,B^2\,(\sqrt{d} - \sqrt{k})^2}$, and $k \ll d$, the last two terms are non-positive, suggesting that SHERPA can yield a no greater generalization upper bound than FT under these conditions.*

*Remark* 3.6. This result provides a bound-level comparison that offers qualitative insight into why restricting updates to a task-relevant subspace (when well estimated and $k \ll d$) may mitigate generalization degradation; it should not be interpreted as a universal guarantee. In practice, we find that $m \ge 50$ is sufficient in our settings; see Appendix M for details.

### 3.2. Overview

Building upon the above theoretical analysis, we design the following methodological pipeline. As illustrated in Figure 2, our method leverages a small SAM $\mathcal{M}_{\text{small}}$ to guide the fine-tuning of a larger SAM $\mathcal{M}_{\text{large}}$ in the high Fisher ratio subspace. Specifically, the method operating by matching the high Fisher Ratio features extracted from both models, $Feat_{\text{task}}^{\text{small}}$ and $Feat_{\text{task}}^{\text{large}}$.

We design a two-stage framework to accomplish the aforementioned task. In the first stage, the input image is passed through the fine-tuned small model. Simultaneously, the GFE module extracts Guiding Feature from the intermediate layers of the mask decoder, and FRS is used to isolate the task-relevant features. In the second stage, the same procedure is applied to extract task-relevant features from the large model. Finally, the more representative task-relevant features from the small model are used to guide the fine-tuning of the large model. Moreover, using a small guiding model further reduces overhead, since self-guidance with the large model would require fine-tuning the large model twice.

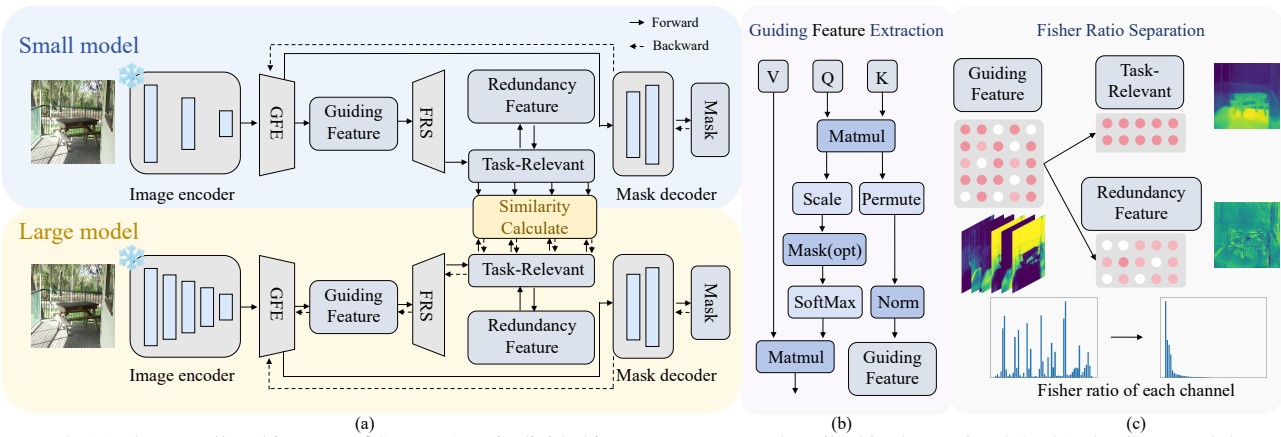

*Figure 2.* (a) The overall architecture of SHERPA. It is divided into two stages, as described in the section 3.2. (b) The GFE Module uses the normalized product of Q and K as the Guiding Feature. (c) The FRS Module. The FRS Module uses an orthogonal transformation that maximizes the Fisher ratio to separate the Task-Relevant features from the Guiding Feature.

## 3.3. Fisher Ratio Separation Module

In our approach, we need to decouple the optimization of task-relevant and generalization-relevant features, confining the fine-tuning process to the task-relevant features. Therefore, the objective of our Fisher Ratio Separation (FRS) is to separate the task-relevant features from the remaining components. However, in practice, this separation is challenging because the task-relevant features are often stochastically distributed across channels, lacking any discernible pattern.

To tackle this issue, we employ the Fisher ratio to isolate task-discriminative directions rather than merely high-response features. By maximizing between-class separation and minimizing within-class variance, it explicitly captures features with the highest task relevance. Control experiments using random subspaces and shuffled labels (Appendix X) confirm our gains stem specifically from this label-aware discriminative subspace.

To achieve this separation, we construct an orthogonal transformation that projects the original features into a subspace. By maximizing the Fisher ratio of the features within this subspace, we obtain the optimal orthogonal transformation matrix. The resulting subspace after projection is the high Fisher ratio subspace, where the task-relevant features with high Fisher ratios are isolated. The remaining features, which are less relevant to the current task but may contain information useful for other generalization tasks, are retained. We then perform fine-tuning only on the task-relevant features. The operation of this orthogonal transformation matrix can be expressed as follows:

$$u = \arg\max_{u \in \mathbb{R}^{c \times k}} \frac{u^\top S_B u}{u^\top S_W u}. \tag{1}$$

Here, $u$ represents the orthogonal transformation we constructed. $k$ is the number of channels in the task-relevant

subspace. $S_B$ and $S_W$ denote the between-class scatter matrix and within-class scatter matrix of Feat, respectively. Feat is the guiding feature extracted by the subsequent GFE, and Feat $\in \mathbb{R}^{c \times d_h}$.

The detailed derivation process is provided in Appendix B. The orthogonal transformation $u$ separates the task-relevant features $Feat_{task}$ from the original guiding features $Attn$ in both $\mathcal{M}_{\text{large}}$ and $\mathcal{M}_{\text{small}}$. We further conduct a Fisher ratio analysis on the separated task-relevant features and the remaining components in the Appendix J.

## 3.4. Guiding Feature Extraction Module

The primary objective of the GFE module is to extract more representative features from the small model. This is because we need to provide a representative task-relevant feature to guide the fine-tuning of the large SAM.

According to the information bottleneck theory, models with limited capacity are compelled to retain only the most task-relevant features. Although smaller SAMs suffer from limited model capacity, which leads to suboptimal fine-tuning performance and generalization, a smaller capacity also imposes a stronger information bottleneck. This stronger bottleneck enables smaller models to obtain a more representative set of high Fisher ratio features during fine-tuning. Moreover, using a small guiding model further reduces overhead, since self-guidance with the large model would require fine-tuning the large model twice.

Unlike previous works, which often adopt the entire module outputs as features, we utilize the normalized product of the query and key matrices as our feature representation. This choice addresses the limitations of using the final module outputs for FRS separation. Final outputs typically undergo multiple layers of nonlinear transformations and multi-head attention, resulting in highly entangled features that mix complex contextual information, thereby hindering effective

separation. In contrast, the product of the query and key matrices provides a more direct and interpretable measure of pixel-level importance. By conducting Fisher ratio analysis on the QK product, we can more clearly assess the contribution of each pixel feature to class discrimination, enabling more precise and effective feature separation. The specific form is as follows:

$$Feat = \text{norm}(QK^T). \tag{2}$$

Before guiding the large model, we first prepare the small model $\mathcal{M}_{\text{small}}$ through a controlled fine-tuning process. In this stage, we optionally apply a KL-divergence regularization between the small model's features during fine-tuning and their pre-trained reference features, which serves as an auxiliary constraint to encourage a tighter information bottleneck. This KL term is not required for SHERPA and only provides an additional improvement. We emphasize that this regularization is used only when training the guiding small model and does not affect the Fisher-ratio subspace construction in FRS.

In summary, for the entire GFE module, it is embedded within the intermediate layers of the mask decoder and extracts the guiding features $Feat_{\text{guiding}}$ from the small model during constrained fine-tuning $\mathcal{M}'_{\text{small}}$. $\mathcal{M}_{\text{small}}$ is fine-tuned on the current task while constraining the original features, allowing the task-relevant features in $Feat_{\text{guiding}}$ to become more representative for the current task.

### 3.5. Loss Functions

Once the task-relevant features $Feat_{task}$ from the small model are extracted, we utilize them to guide the large model's task-relevant feature within the same subspace. This process is different from traditional knowledge distillation, which usually matches predictions or global features between models. Instead, we explicitly align only the task-relevant subspace, focusing the transfer on essential information and preserving features related to generalization. This leads to a different transfer mechanism that better preserves generalization-related representations in practice.

Specifically, the task-relevant feature from the small model serves as a supervisory signal, encouraging the large model to focus on similar task-critical regions. This guidance is implemented via an alignment loss based on the $L_1$-distance between the two task-relevant features:

$$\mathcal{L}_{\text{guiding}} = \sum_{i=1}^{k} \left| Feat_{\text{task},i}^{large} - Feat_{\text{task},i} \right|. \tag{3}$$

$Feat_{\text{task}}^{large}$ represents the task-relevant features extracted from the large model using the above method. By confining the fine-tuning process to task-relevant features, this approach minimizes interference with generalization-related

features, thereby better preserving the model's generalization ability. Meanwhile, to ensure the accuracy of the model's output, we also include a task-specific loss. The final loss function thus combines this alignment loss with a task-specific loss, such as mean squared error:

$$\mathcal{L}_{\text{total}} = \lambda \cdot \mathcal{L}_{\text{guiding}} + \mathcal{L}_{\text{task-specific}}(y_{\text{pred}}, y_{\text{true}}), \tag{4}$$

where $\lambda$ balances feature alignment and task performance, ensuring improved model effectiveness. We have performed ablation on this parameter in Appendix K.

## 4. Experiments

### 4.1. Datasets and Metrics

**Datasets.** To achieve noticeable performance improvements, we select the DISK-5k (Qin et al., 2022) and DUTS (Wang et al., 2017) datasets for fine-tuning SAM in the natural image segmentation task. DISK-5k is a dataset designed for highly accurate object segmentation, focusing on targets with varied structural complexities. Previous research has shown that SAM struggles with DISK-5K type datasets (Ke et al., 2024) while facing less of a challenge with the DUTS dataset.

Fine-tuning a model pre-trained on natural images, such as SAM, for biomedical images has been a valuable area of research (Wu et al., 2023; Cai et al., 2024). Thus, we select the Lucchi dataset (Lucchi et al., 2013), a biomedical image segmentation dataset specifically designed for mitochondrial segmentation in electron microscopy images. For the video segmentation task, we select the VOST (Tokmakov et al., 2023) dataset to fine-tune SAM2.

Additionally, we select twelve other datasets to assess the model's generalization capability. The datasets used to test generalization performance on natural image segmentation include the following six: ADE20K (Xia et al., 2019), Cityscapes (Garcia-Garcia et al., 2017), COCO-stuff (Anwar et al., 2020), ECSSD (Tran et al., 2020), FSS (Li et al., 2020b), BIG (Cheng et al., 2020). Among these, the first three datasets, which are significantly different from the fine-tuning dataset, are categorized as Group 1, while the latter three, which are more similar, are categorized as Group 2. The datasets used to test segmentation on biomedical images include the following three: VNCIII, MitoEM-R (Wei et al., 2020), and MitoEM-H (Wei et al., 2020). The datasets used for video segmentation include the following three: UVO (Wang et al., 2021), VIPSeg (Miao et al., 2022), and PUMaVOS (Bekuzarov et al., 2023). In addition, we incorporate more diverse segmentation tasks, such as part segmentation and background segmentation.

**Metrics.** For natural image segmentation, We use the instance-level F1 score (F1), mean Intersection over Union (mIoU), and pixel-level Dice score (Dice). For biomedical

*Table 1.* Natural Image Segmentation Performance Comparison Across Datasets. This table summarizes the performance metrics across various datasets and methods. The **Valid** column represents the performance on the specific task validation set. **Group 1** represents generalization datasets with significant differences from the fine-tuning dataset, and **Group 2** represents datasets that are more similar to the fine-tuning dataset. The **Average** column reflects the overall average generalization performance. **Retention** represents the model's retention of generalization capability, and we set the zero-shot performance as the baseline at 100%. The numbers after $L^2$-SP and Ft-last represent different settings. For specific settings, see the Appendix E. Citys indicates the Cityscapes dataset.

| Fine-tuning Dataset | Method | Valid | Group 1 | | | Group 2 | | | Average | Retention |
|---|---|---|---|---|---|---|---|---|---|---|
| | | | ADE20K | Citys | COCO | ECSSD | FSS | BIG | | |
| | Zero-shot | 0.6570 | 0.8552 | 0.7335 | 0.6921 | 0.9293 | 0.9434 | 0.9363 | 0.8483 | 100% |
| DISK-5k | Std-ft | 0.8728 | 0.5062 | 0.4390 | 0.6186 | 0.9405 | 0.9151 | 0.9222 | 0.7236 | 85.3% |
| | KL-SP | 0.8621 | 0.6012 | 0.5581 | 0.6732 | 0.9513 | 0.9131 | 0.7512 | 0.7414 | 87.4% |
| | $L^2$-SP3 | 0.8571 | 0.5201 | 0.4710 | 0.6202 | 0.9620 | 0.9370 | 0.8709 | 0.7302 | 86.1% |
| | Ft-last4 | 0.8495 | 0.6468 | 0.5387 | 0.6730 | 0.9538 | 0.9344 | 0.9121 | 0.7764 | 91.5% |
| | FisherTune | 0.8732 | 0.6242 | 0.5643 | 0.6334 | 0.9344 | 0.9070 | 0.9031 | 0.7594 | 89.5% |
| | InfoSAM | 0.8834 | 0.6398 | 0.5485 | 0.6842 | 0.9541 | 0.9361 | 0.8765 | 0.7732 | 91.1% |
| | Ours | **0.8855** | **0.6553** | **0.5729** | **0.6917** | **0.9643** | **0.9407** | **0.9331** | **0.7930** | **93.4%** |
| | Zero-shot | 0.8930 | 0.8552 | 0.7335 | 0.6921 | 0.9493 | 0.9434 | 0.9363 | 0.8483 | 100% |
| DUTS | Std-ft | 0.9402 | 0.5857 | 0.5743 | 0.5931 | 0.9547 | 0.9185 | 0.9188 | 0.7575 | 89.3% |
| | KL-SP | 0.9364 | 0.7431 | 0.6012 | 0.6341 | 0.9451 | 0.9173 | 0.8631 | 0.7840 | 92.4% |
| | $L^2$-SP3 | 0.9231 | 0.6411 | 0.5913 | 0.5832 | 0.9515 | 0.9245 | 0.8482 | 0.7566 | 89.1% |
| | Ft-last4 | 0.9394 | **0.7676** | 0.6478 | 0.6776 | 0.9210 | 0.9280 | 0.9319 | 0.8123 | 95.8% |
| | FisherTune | 0.9346 | 0.6877 | 0.6215 | 0.6303 | 0.9461 | 0.9160 | 0.8592 | 0.7769 | 91.5% |
| | InfoSAM | 0.9465 | 0.7247 | 0.6417 | 0.6522 | 0.9496 | 0.9294 | 0.9162 | 0.8023 | 94.5% |
| | Ours | **0.9495** | 0.7672 | **0.6810** | **0.6817** | **0.9682** | **0.9369** | **0.9355** | **0.8284** | **97.7%** |

image segmentation, the F1 score is applied. For video segmentation, we adopt the $\mathcal{J}\&\mathcal{F}$ metric, where $\mathcal{J}$ represents the mIoU between the predicted mask and the ground truth, and $\mathcal{F}$ measures the alignment between the boundaries of the predicted mask and the ground truth boundaries. Detailed metric calculations and additional dataset information are provided in the Appendix E.

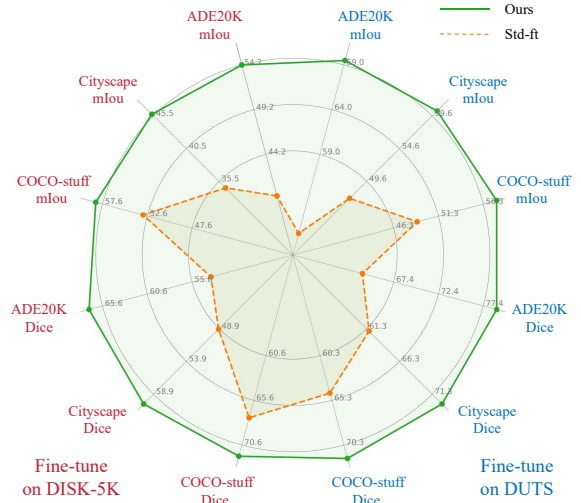

*Figure 3.* The radar chart summarizes generalization performance after standard and SHERPA fine-tuning.

Additionally, to assess the model's retention of generaliza-

tion ability, we set the zero-shot performance as the baseline at 100%. The generalization ability retention of other models is then calculated as a ratio compared to the zero-shot performance.

### 4.2. Experiments Setting

In natural image segmentation, we use SAM (Kirillov et al., 2023) vit-h as the large model requiring fine-tuning and SAM vit-b as the guiding small model. Notably, the large model outperforms the small model in both zero-shot and standard fine-tuning settings.

In the main paper, we use a box as the prompt and evaluate only the first mask output. Additional results using alternative prompts and evaluating other mask outputs are provided in the Appendix C. Details on the settings for Natural image segmentation, Biomedical Image Segmentation, and Video Segmentation Performance can be found in the Appendix G.

We experiment with various robust fine-tuning methods. Ultimately, we select five baseline methods: Zero-shot, where the model is evaluated directly on new data without fine-tuning; Std-ft, which fine-tunes the model's mask decoder with consistent hyperparameter settings; Ft-last (Wortsman et al., 2022), which fine-tunes only the last few layers of the model; and $L^2$-SP (Li et al., 2018), a method proposed to mitigate generalization loss through a quadratic penalty.

KL-SP, where we also apply the KL-divergence constraint to fine-tune large models, is included as a baseline. We also include FisherTune (Zhao et al., 2025) and InfoSAM (Zhang et al., 2025), two strong methods that aim to preserve useful information during fine-tuning via Fisher-based parameter protection and information-preserving regularization, respectively. Further details on the baselines and additional baseline settings are provided in Appendix L.

To further validate the effectiveness of our method, in addition to standard fine-tuning, we also experiment with other tuning strategies, such as LoRA-based (Hu et al., 2022) and adapter-based methods. These results are presented in the Appendix D. Results on more diverse segmentation tasks, such as part segmentation and background segmentation, are also included in the Appendix C.

### 4.3. Results and Analysis

Our natural image segmentation results are shown in Table 1 and Figure 1. Biomedical image segmentation results are shown in Table 2, all metrics in this table are instance-level F1 scores. Table 3 shows video segmentation results, measured using the $\mathcal{J}\&\mathcal{F}$ score. Our method improves specific task performance while simultaneously reducing generalization loss, outperforming all baselines across four fine-tuning datasets and twelve generalization datasets used in the three tasks.

**Generalization Retention.** Based on the generalization performance metrics of zero-shot and standard fine-tuning across multiple datasets, we can observe that the model's generalization ability tends to decline after fine-tuning. At the same time, this decline is related to the similarity between the generalization datasets and the fine-tuning dataset. The less similar the generalization dataset is to the fine-tuning dataset, the more pronounced the decline becomes. Compared to previous approaches, our method is more effective in mitigating the loss of generalization capability while maintaining strong task-specific performance. The Figure 4 demonstrates the generalization performance of our method on biomedical image segmentation. Moreover, we visualize the generalization results for biomedical image segmentation in Appendix O.

**Specific task Performance.** Our method alleviates the loss of generalization ability while slightly improving specific task performance compared to standard fine-tuning approaches. Specifically, our method improves F1 scores by 1.1% on natural image segmentation tasks; increases F1 scores by 1.1% on biomedical image segmentation; and achieves a 2.3% improvement in $\mathcal{J}\&\mathcal{F}$ on video segmentation. As a comparison, traditional methods (such as KL-SP, L2-SP3, and Ft-last4) for mitigating generalization loss often do so by sacrificing task-specific performance. Specifi-

*Table 2.* Biomedical Image Segmentation Performance. The **Natural** column reflects the model's average generalization performance on natural image datasets. All metrics in the table are instance-level F1 scores.

| Method | Valid | Dataset | | | Retention | Natural |
|---|---|---|---|---|---|---|
| | | VNCIII | Mito-R | Mito-H | | |
| Zero-shot | 0.872 | 0.934 | 0.820 | 0.812 | 100% | 0.848 |
| Std-ft | 0.911 | 0.904 | 0.571 | 0.550 | 78.9% | 0.416 |
| KL-SP | 0.852 | 0.723 | 0.523 | 0.573 | 70.9% | 0.514 |
| $L^2$-SP3 | 0.828 | 0.819 | 0.516 | 0.510 | 71.9% | 0.567 |
| Ft-last4 | 0.902 | 0.912 | 0.581 | 0.567 | 80.3% | 0.705 |
| FisherTune | 0.882 | 0.896 | 0.592 | 0.628 | 82.5% | 0.674 |
| InfoSAM | 0.919 | **0.929** | 0.601 | 0.692 | 86.6% | 0.694 |
| Ours | **0.923** | 0.919 | **0.714** | **0.724** | **91.8%** | **0.754** |

*Table 3.* Video Segmentation Performance. During training, we used a fixed set of 16 frames, while during testing, we evaluated all available frames. All metrics in the table are $\mathcal{J}\&\mathcal{F}$.

| Method | Valid | Dataset | | | Retention |
|---|---|---|---|---|---|
| | | UVO | VIPSeg | PMVOS | |
| Zero-shot | 0.450 | 0.668 | 0.545 | 0.542 | 100% |
| Std-ft | 0.555 | 0.513 | 0.492 | 0.470 | 84.0% |
| KL-SP | 0.532 | 0.503 | 0.483 | 0.492 | 84.2% |
| $L^2$-SP3 | 0.526 | 0.502 | 0.472 | 0.491 | 83.5% |
| Ft-last4 | 0.548 | 0.563 | 0.511 | 0.497 | 89.6% |
| FisherTune | 0.539 | 0.495 | 0.476 | 0.512 | 83.4% |
| InfoSAM | 0.564 | 0.558 | 0.476 | 0.494 | 87.1% |
| Ours | **0.578** | **0.642** | **0.515** | **0.512** | **95.1%** |

cally, we experiment with multiple configurations of $L^2$-SP, selecting $L^2$-SP3 and $L^2$-SP4 for their best fine-tuning results. However, on the DISK-5K task, they cause a 6.79% drop in task-specific performance. Similarly, using Ft-last4 and Ft-last2, resulted in a 7.17% decrease in task-specific performance. The visualization results are shown in the Appendix O.

**Cross-domain Retention.** Another notable improvement is in cross-domain generalization retention. Typically, a model pre-trained on natural images experiences a greater drop in generalization ability after fine-tuning on biomedical images. For instance, after fine-tuning on the Lucchi dataset, the model's performance on natural image datasets declines significantly. This is primarily due to the substantial distributional differences between biomedical and natural image data. However, as shown in Table 2, our approach effectively mitigates this loss of generalization ability, significantly preserving cross-domain performance.

### 4.4. Computational Overhead and Method Extension

While the guiding model increases training FLOPs by 18%, SHERPA introduces zero inference overhead. For practical deployments (e.g., interactive annotation tools) requiring a single model for both specialized tasks and general scenes,

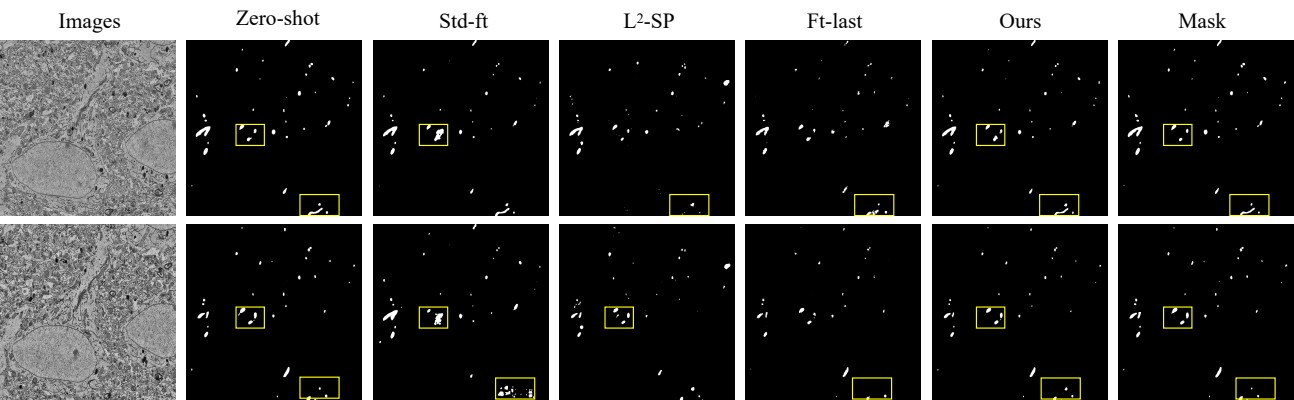

*Figure 4.* Visualization of generalization results on MitoEM-R after fine-tuning on Lucchi.

*Table 4.* Ablation Study for Guiding Model Configuration in DISK-5K. All metrics in the table are instance-level F1 scores.

| Guiding Model | Regularization | Guidance | Valid | General |
|---|---|---|---|---|
| Small | ✓ | ✗ | 0.8502 | 0.7431 |
| Small | ✓ | ✓ | 0.8618 | 0.7466 |
| Large | ✗ | ✓ | 0.8801 | 0.7797 |
| Small (w/o KL) | ✗ | ✓ | 0.8745 | 0.7812 |
| Small | ✗ | ✓ | **0.8855** | **0.7930** |

*Table 5.* Ablation study for FRS module. All metrics are F1.

| Dataset | Method | Valid | General |
|---|---|---|---|
| DISK-5K | w/o FRS | 0.8695 | 0.5752 |
| | with FRS | **0.8855** | **0.7930** |
| DUTS | w/o FRS | 0.9327 | 0.6432 |
| | with FRS | **0.9495** | **0.8284** |
| Lucchi | w/o FRS | 0.9104 | 0.7611 |
| | with FRS | **0.9227** | **0.7857** |

this one-time training cost is a worthwhile trade-off to preserve zero-shot capabilities. Training overhead is further minimized by the lightweight FRS module (Appendix I) and an optional efficiency variant (Appendix C.4). Finally, SHERPA extends to other ViT-based architectures like DINOv2 and MaskFormer (Appendix H).

### 4.5. Ablation Study

**Guiding Model Configuration.** We conducted an ablation study on the design choice of using a constrained fine-tuned small model's task-relevant feature to guide the large model. Specifically, we compared the effectiveness of regularizing the generalization-relevant components versus fine-tuning with task-relevant feature guidance. Additionally, we investigated whether to use the small model for guidance or rely on the large model itself, as well as whether the small model should be fine-tuned with KL divergence constraints. The results, shown in Table 4, provide insights into the effectiveness of these configurations. Using the fine-tuned large model itself for guidance can achieve slightly worse performance than using a small guiding model, but it requires fine-tuning the large model twice, resulting in

*Table 6.* Ablation study for different Var Ratio. All results are from fine-tuning on DISK-5K.

| Var Ratio | Valid | | General | |
|---|---|---|---|---|
| | mIoU | F1 | mIoU | F1 |
| 0.00 | 0.8043 | 0.8789 | 0.6531 | 0.7234 |
| 0.25 | 0.8101 | 0.8816 | 0.7031 | 0.7531 |
| 0.50 | 0.8129 | 0.8850 | 0.6994 | 0.7791 |
| 0.75 | **0.8143** | **0.8855** | **0.7212** | **0.7930** |
| 1.00 | 0.7908 | 0.8695 | 0.4731 | 0.5257 |

much higher overhead ($\sim 1.6\times$) than small-model guidance. Directly regularizing the residual components separated by FRS is suboptimal, as these components do not exclusively contain generalization-related information; they may also introduce noise, which could interfere with the model's generalization ability and degrade task-specific performance. In addition, we provide further parameter sensitivity analysis in Appendix M and robustness evaluations in the Appendix N.

**FRS module.** We assess the effectiveness of the FRS module on the image segmentation datasets. We compare the performance of fine-tuning the large model using the guiding small model directly versus fine-tuning with the guiding small model based on the FRS module. As shown in Table 5, using the FRS module reduces the degradation of generalization ability and improves performance on the fine-tuning datasets.

**Number of Task-Relevant Feature Channels.** The number of channels for the original features extracted by different models is not fixed. We use the proportion of the Fisher ratio of the task-relevant feature channels to the total Fisher ratio of the original guiding feature as a selection criterion. We consider 5 scenarios: 0.00, 0.25, 0.50, 0.75, and 1.00. As shown in Table 6, the 0.75 setting yields the best performance. Fewer channels are insufficient to separate task-relevant features, while more channels introduce excessive generalization redundancy. The 0.75 ratio corresponds to the first three channels for SAM and ten for SAM2.

# 5. Conclusion

We propose a fine-tuning method in which a small SAM guides the large SAM, named SHERPA. By extracting task-relevant features from the small SAM to guide fine-tuning of the large SAM, SHERPA reduces the degradation of generalization ability and improves task-specific performance.

# Acknowledgements

This work was supported in part by the National Natural Science Foundation of China under Grant 624B2137.

# Impact Statement

This paper presents a method to improve fine-tuning of Segment Anything Models for specialized segmentation tasks while better retaining general-purpose capability. Such improvements may reduce the cost of adapting segmentation systems across domains and applications. Potential risks include privacy-invasive uses of stronger segmentation and harmful errors under distribution shift in high-stakes settings; we recommend domain-specific validation, clear reporting of limitations, and appropriate oversight for sensitive deployments.

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

## A. Theoretical Analysis on Sherpa

### A.1. Preliminaries and Notation

**Assumption A.1.** Finite Second Moment For $x \sim \mathcal{D}$,

$$\mathbb{E} \left\| small f(x) \right\|_2^2 \leq B^2, \qquad \mathbb{E} \left\| large f^0(x) \right\|_2^2 \leq B^2.$$

**Assumption A.2.** High Fisher Ratio Subspace. Let $k < d$ be given. Using *only* the first $m$ samples $\{x_i, y_i\}_{i=1}^m$ we estimate

$$W = \underset{W^\top W = I_k}{\arg\max} \; FR\big(W^\top small f(x); y\big),$$

$$\bar{W} : \text{Orthogonal complement matrix.}$$

Throughout the generalisation analysis, $W$ is treated as fixed; the price of Data dependence is quantified by an extra term $\Delta_W$ in Theorem A.5.

### A.2. Training Objectives (second split of size $n$)

We focus on *mean-squared error* (MSE) for clarity.

**Standard fine-tuning (FT).**

$$\hat{R}_{FT}(\theta) = \frac{1}{n} \sum_{i=1}^n \big(y_i - g(large f_\theta(x_i))\big)^2.$$

**SHERPA fine-tuning.**

$$\hat{R}_{SHERPA}(\theta) = \frac{1}{n} \sum_{i=1}^n \big(y_i - g(large f_\theta(x_i))\big)^2$$
$$+ \lambda \frac{1}{n} \sum_{i=1}^n \big\| W^\top large f_\theta(x_i) - W^\top small f(x_i) \big\|_1 \tag{5}$$

### A.3. $\ell_1$–$\ell_2$ Decomposition

For any $v \in \mathbb{R}^d$,

$$\|v\|_1 \leq \|W^\top v\|_1 + \|\bar{W}^\top v\|_1.$$

We call $L_{\text{task}}(\theta) = \frac{1}{n} \sum_i \|W^\top (large f_\theta - small f)(x_i)\|_1$.

**Lemma A.3** (Approximate optimality). *Let $\theta^\star$ minimise equation 5. Then for any $\beta > 0$*

$$L_{task}(\theta^\star) \leq \varepsilon(\lambda)$$

*where $\varepsilon(\lambda) \downarrow 0$ as $\lambda \uparrow \infty$.*

### A.4. Capacity Control

Assume the final segmentation head $g$ belongs to a class $\mathcal{G}$ whose empirical Rademacher complexity is upper-bounded by $C_g$; the feature extractor is $L_f$-Lipschitz in parameters.

**Lemma A.4** (Rademacher Complexity). *Let $\mathcal{H}_{FT}$ (resp. $\mathcal{H}_{SHERPA}$) be the hypothesis set reachable by FT (resp. SHERPA) after the second split of size $n$. Then*

$$\mathcal{R}_n(\mathcal{H}_{FT}) = O\big(C_g B \sqrt{d/n}\big),$$

$$\mathcal{R}_n(\mathcal{H}_{SHERPA}) = O\big(C_g B \sqrt{k/n}\big).$$

*Sketch.* only $k$ coordinates of the feature vector are *trainable* in SHERPA, the remaining $d - k$ being fixed constants. Full details are deferred to Appendix A. $\qquad\square$

Let $\text{GenGap}(\cdot)$ denote the usual Rademacher generalisation bound. Lemma A.4 directly yields

$$\text{GenGap}_{SHERPA} \leq \text{GenGap}_{FT} \sqrt{k/d}. \tag{1}$$

### A.5. Main Result

**Theorem A.5** (Risk upper-bound: SHERPA $\leq$ FT). *Use $m$ samples for subspace estimation and $n$ for training, $m + n = N$. Assume $k \ll n \ll d$ and choose hyper-parameters $\lambda, \beta$ are large enough such that the conditions in Lemma A.3 holds. Then with probability at least $1 - \delta$*

$$R(\theta_{\text{SHERPA}}) \leq R(\theta_{\text{FT}}) + \underbrace{\tilde{\Delta}_W}_{\text{subspace selection}} + \underbrace{\frac{C\, C_w^2 B^2}{\sqrt{n}} \left(\sqrt{k} - \sqrt{d}\right)}_{\text{capacity gain}}$$

*where $R(\cdot)$ denotes an upper bound on the generalization risk of the model, and $\tilde{\Delta}_W = O(B\sqrt{k \log d/m})$ quantifies the subspace estimation error. In particular, if $m \geq \frac{a^2\, n\, k\, \log(d/\delta)}{C^2\, C_w^4\, B^2\, (\sqrt{d} - \sqrt{k})^2}$, and $k \ll d$, the last two terms are non-positive, suggesting that SHERPA can yield a no-larger generalization upper bound than FT under these conditions.*

*Proof.* For MSE we have the classic decomposition Risk = Bias + GenGap. The bias difference is controlled via Lemma A.3; the gap difference uses equation 1. Add the model selection penalty $\Delta_W$ induced by sample–splitting (Appendix B). $\qquad\square$

## B. FRS Details.

In this section, we provide strict mathematical proof for the design of the Feature Redundancy Separation (FRS) module.

### B.1. Diagonalizability Assumption and Fisher Ratio–Variance Equivalence

The Fisher ratio for a direction $u$ is defined as:

$$FR(u) = \frac{u^\top \Sigma_b u}{u^\top \Sigma_w u}$$

where $\Sigma_b$ and $\Sigma_w$ are the between-class and within-class covariance matrices, respectively.

**Key theoretical fact:** If $\Sigma_b$ and $\Sigma_w$ are *simultaneously diagonalizable* (i.e., there exists an orthonormal basis where both are diagonal), then maximizing the Fisher ratio in a $k$-dimensional subspace is equivalent to maximizing the total variance in that subspace (assuming the within-class scatter has been normalized, e.g., by whitening). In our context, after mean-centering and normalization (e.g., via Layer-Norm), the within-class scatter $\Sigma_w$ becomes approximately isotropic, so the simultaneous diagonalizability condition is approximately satisfied.

**Empirical verification:** We directly measure the commutation error between $\Sigma_b$ and $\Sigma_w$ as

$$\rho_1 = \frac{\|\Sigma_b \Sigma_w - \Sigma_w \Sigma_b\|_F}{\|\Sigma_b\|_F \|\Sigma_w\|_F}.$$

In our experiments on the SAM backbone, we observe $\rho_1 \approx 0.04$. In the literature, $\rho_1 < 0.06$ is widely considered "approximately commuting" and thus "approximately simultaneously diagonalizable."

Therefore, maximizing the Fisher ratio can be reduced to maximizing the variance.

### B.2. Assumptions.

- $Feat \in \mathbb{R}^{m \times d}$: The guiding feature, where $m$ is the number of channels in the guiding feature, and $d$ is the number of feature dimensions.

- $k$: The number of channels in the task-relevant subspace ($k \leq m$).

- $u \in \mathbb{R}^{m \times k}$: The projection matrix we aim to find.

- The feature $Feat$ is mean-centered, i.e., each channel has zero mean:

$$\frac{1}{d} \sum_{j=1}^{d} Feat_{:,j} = 0. \tag{6}$$

### B.3. Objective Function.

Our goal is to find a projection matrix $u$ that maximizes the variance of the projected features in the task-relevant subspace. The optimization problem is formulated as:

$$\max_{u \in \mathbb{R}^{m \times k}} \sum_{i=1}^{k} \sum_{j=1}^{d} \left( u_{:,i}^T Feat_{:,j} \right)^2. \tag{7}$$

### B.4. Derivation

**Matrix Formulation.** The objective function can be expressed in matrix form:

$$\sum_{i=1}^{k} \sum_{j=1}^{d} \left( u_{:,i}^T Feat_{:,j} \right)^2 = \left| u^T Feat \right|_F^2, \tag{8}$$

where $| \cdot |_F$ denotes the Frobenius norm.

**Trace Representation.** Expanding the Frobenius norm yields:

$$\left| u^T Feat \right|_F^2 = \text{tr} \left( u^T Feat Feat^T u \right). \tag{9}$$

**Eigenvalue Decomposition.** Let $A = Feat Feat^T \in \mathbb{R}^{m \times m}$, which is a symmetric positive semi-definite matrix. Then, $A$ can be decomposed as:

$$A = V \Lambda V^T, \tag{10}$$

where:

- $V \in \mathbb{R}^{m \times m}$ is an orthogonal matrix whose columns are the eigenvectors of $A$.

- $\Lambda = \text{diag}(\lambda_1, \lambda_2, \ldots, \lambda_m)$ contains the eigenvalues of $A$ in descending order:

$$\lambda_1 \geq \lambda_2 \geq \cdots \geq \lambda_m \geq 0. \tag{11}$$

**Transformation.** Substituting $A$ into equation 9, we have:

$$\text{tr} \left( u^T A u \right) = \text{tr} \left( u^T V \Lambda V^T u \right). \tag{12}$$

Let us define:
$$W = V^T u. \tag{13}$$
Since $V$ is orthogonal ($V^T V = V V^T = I$), any $u$ can be represented as $u = VW$.

**Simplifying the Trace.** Substituting $u = VW$ into equation 12:

$$\text{tr} \left( u^T V \Lambda V^T u \right) = \text{tr} \left( W^T \Lambda W \right). \tag{14}$$

**Maximization.** Since $\Lambda$ is a diagonal matrix, equation 14 becomes:

$$\text{tr} \left( W^T \Lambda W \right) = \sum_{i=1}^{k} \lambda_i \left| W_{i,:} \right|^2. \tag{15}$$

To maximize this sum, we need to select $W$ such that it aligns with the largest eigenvalues $\lambda_i$. The maximum is achieved when:

$$W = \begin{bmatrix} I_k & 0 \end{bmatrix}, \tag{16}$$

where $I_k$ is the $k \times k$ identity matrix, and 0 is a $(m-k) \times k$ zero matrix.

**Optimal Projection Matrix.** Therefore, the optimal $u$ is:

$$u = VW = V\begin{bmatrix} I_k & 0 \end{bmatrix} = V_{:,1:k}. \qquad (17)$$

The columns of $u$ are the eigenvectors corresponding to the largest $k$ eigenvalues of $A$. By projecting onto these eigenvectors, we maximize the variance in the task-relevant subspace, effectively separating task-relevant features from redundant ones.

**Remark.** This derivation shows that the FRS module extracts the most significant features that capture the maximum variance, which is assumed to be most relevant to the task.

## C. Additional Evaluation Results

### C.1. More Prompt Evaluation

SAM supports various types of prompts. To further assess the robustness of our method under varied input conditions, we conduct additional evaluations using diverse types of prompts. While the main paper focuses on box prompts, here we extend the evaluation to point prompts and mask prompts. For point prompts, we use ten points: one located at the center of mass of the ground-truth mask, and the other nine selected randomly within the mask region. For mask prompts, we use degraded coarse masks that simulate imperfect annotations. The results are shown in Table 7 and 8, demonstrating that our method consistently performs well under these alternative prompting conditions.

### C.2. More Output Mask Evaluation

SAM has multiple outputs. To better evaluate the effectiveness of our method, we test multiple output masks. In the main paper, we only report results based on the first output mask. However, SAM generates multiple output masks for each input. In this section, we evaluate the performance of the top three output masks. The results are presented in

*Table 7.* Evaluation results for each task under point-based prompting. All metrics are reported as F1 scores.

| Dataset | Method | Valid | General |
|---------|--------|-------|---------|
| DISK-5K | Zero-shot | 0.6321 | 0.8028 |
|         | Std-ft | 0.8623 | 0.7102 |
|         | SHERPA | 0.8734 | 0.7812 |
| DUTS    | Zero-shot | 0.8892 | 0.8502 |
|         | Std-ft | 0.9421 | 0.7672 |
|         | SHERPA | 0.9502 | 0.8314 |
| Lucchi  | Zero-shot | 0.8523 | 0.8203 |
|         | Std-ft | 0.9123 | 0.6841 |
|         | SHERPA | 0.9125 | 0.7634 |

*Table 8.* Evaluation results for each task under mask-based prompting. All metrics are reported as F1 scores.

| Dataset | Method | Valid | General |
|---------|--------|-------|---------|
| DISK-5K | Zero-shot | 0.6512 | 0.8341 |
|         | Std-ft | 0.8712 | 0.7467 |
|         | SHERPA | 0.8789 | 0.7821 |
| DUTS    | Zero-shot | 0.8701 | 0.8513 |
|         | Std-ft | 0.9432 | 0.7612 |
|         | SHERPA | 0.9502 | 0.8311 |
| Lucchi  | Zero-shot | 0.8752 | 0.8443 |
|         | Std-ft | 0.9214 | 0.6884 |
|         | SHERPA | 0.9278 | 0.7924 |

*Table 9.* Evaluation results for the second output mask. All metrics are reported as F1 scores.

| Dataset | Method | Valid | General |
|---------|--------|-------|---------|
| DISK-5K | Zero-shot | 0.6672 | 0.8391 |
|         | Std-ft | 0.8728 | 0.7324 |
|         | SHERPA | 0.8915 | 0.8102 |
| DUTS    | Zero-shot | 0.8992 | 0.8533 |
|         | Std-ft | 0.9482 | 0.7655 |
|         | SHERPA | 0.9552 | 0.8342 |
| Lucchi  | Zero-shot | 0.8765 | 0.8612 |
|         | Std-ft | 0.9215 | 0.6821 |
|         | SHERPA | 0.9314 | 0.7912 |

Table 9 and Table 10, showing how our method performs across multiple candidate outputs.

### C.3. Background and Partial Segmentation Evaluation

As SAM is a foundation model, it is crucial to evaluate its performance across a broader range of tasks. Therefore, we additionally report results on background segmentation in salient object detection tasks, as well as partial segmentation performance on the Pascal VOC dataset. The results are summarized in Table 11.

### C.4. Computational Overhead Analysis

Moreover, we can further reduce the time cost by appropriately decreasing the number of training epochs for the small model, reducing the training data for the small model, or choosing an even smaller model (such as the pruned slim-SAM, which has only 26M parameters). The following table presents a comparison between computational time and performance.

The extra computational overhead in our method mainly comes from training the small model. Therefore, by using fewer training epochs, a smaller training dataset, or an even

*Table 10.* Evaluation results for the third output mask. All metrics are reported as F1 scores.

| Dataset | Method | Valid | General |
|---|---|---|---|
| DISK-5K | Zero-shot | 0.6642 | 0.8402 |
|  | Std-ft | 0.8736 | 0.7421 |
|  | SHERPA | 0.8932 | 0.8153 |
| DUTS | Zero-shot | 0.8972 | 0.8231 |
|  | Std-ft | 0.9321 | 0.7203 |
|  | SHERPA | 0.9462 | 0.8123 |
| Lucchi | Zero-shot | 0.8765 | 0.8612 |
|  | Std-ft | 0.9311 | 0.6841 |
|  | SHERPA | 0.9423 | 0.7812 |

*Table 11.* Evaluation results for the second output mask. All metrics are reported as F1 scores.

| Dataset | Method | Valid | General |
|---|---|---|---|
| Background | Zero-shot | 0.4125 | 0.8102 |
|  | Std-ft | 0.6654 | 0.7102 |
|  | SHERPA | 0.6874 | 0.7832 |
| Partial | Zero-shot | 0.7231 | 0.8533 |
|  | Std-ft | 0.8293 | 0.7421 |
|  | SHERPA | 0.8392 | 0.8412 |

*Table 12.* Performance of Combination with LoRA and adapter in DISK-5k.

| Setting | Method | Valid | General |
|---|---|---|---|
| LoRA-Encoder | w/o SHERPA | 0.9216 | 0.6781 |
|  | with SHERPA | 0.9286 | 0.7364 |
| LoRA-Decoder | w/o SHERPA | 0.8638 | 0.7911 |
|  | with SHERPA | 0.8693 | 0.8193 |
| Adapter-Encoder | w/o SHERPA | 0.9121 | 0.6643 |
|  | with SHERPA | 0.9293 | 0.7423 |

*Table 13.* Trade-off between computational overhead and performance for different small-model settings. "Valid" and "Generalization" denote the in-distribution and cross-dataset performance, respectively.

| Small Model Setting | Valid | Generalization | Overhead |
|---|---|---|---|
| std-ft (no guide) | 0.8728 | 0.7236 | 11.7 GPU hours |
| 1/2 epochs | 0.8810 | 0.7824 | 14.6 GPU hours |
| 1/4 epochs | 0.8745 | 0.7731 | 13.4 GPU hours |
| 1/2 data | 0.8785 | 0.7692 | 14.8 GPU hours |
| 1/4 data | 0.8749 | 0.7623 | 13.5 GPU hours |
| slimSAM | 0.8842 | 0.7924 | 12.8 GPU hours |
| full | 0.8855 | 0.7930 | 15.6 GPU hours |

smaller small model, we can further reduce the time cost. Importantly, as shown in Table 13, these modifications still maintain comparable performance. Notably, due to the stronger bottleneck effect in slimSAM, it achieves similar results to the original ViT-B model while requiring even less computational time (only 9% of the original pre-training cost).

## D. Additional Fine-Tuning Adaptation Strategies

In addition to standard fine-tuning, LoRA-based and adapter-based fine-tuning are also widely adopted approaches in transfer learning. To validate the generality and flexibility of our method, beyond the standard fine-tuning adaptation used in the main paper, we further evaluate LoRA-based and adapter-based fine-tuning strategies. The corresponding results are presented in Table 12. We observe that applying LoRA to fine-tune the encoder of SAM leads to notable improvements in task-specific performance. However, this gain comes at the cost of a significant reduction in generalization ability. In contrast, our method effectively alleviates this degradation while further boosting fine-tuning performance. When LoRA is applied to the decoder, the performance changes are relatively marginal, yet our method continues to provide improvements, demonstrating its adaptability across various fine-tuning configurations.

## E. Dataset Details.

To achieve noticeable performance changes, we selected the DISK-5k (Qin et al., 2022) and DUTS (Wang et al., 2017) datasets for fine-tuning for natural image segmentation. DISK-5k is a dataset designed for highly accurate object segmentation, focusing on targets with varied structural complexities. It contains 5,470 images across 22 groups and 225 categories, with pixel-wise labeling to ensure precision. Previous studies have shown that SAM struggles with these types of datasets (Ke et al., 2024). DUTS is a saliency detection dataset containing 10,553 training images and 5,019 test images. All training images are collected from the ImageNet DET training/val sets, while test images are collected from the ImageNet DET test set and the SUN data set. It is less challenging for SAM. Table 14 provides detailed information about the datasets.

The imaging modalities of biomedical images and natural images exhibit significant differences, leading to substantial variations in data distribution. Fine-tuning a model pre-trained on natural images, such as SAM, for biomedical images has been a valuable area of research(Wu et al., 2023; Cai et al., 2024). Thus, adapting the model to biomedical images while retaining its generalization capability from natural images presents a meaningful and challenging problem. We selected the Lucchi (Lucchi et al., 2013) dataset. is a biomedical image segmentation dataset specifically designed for mitochondrial segmentation in electron microscopy images. It includes annotated sub-volumes taken from the CA1 hippocampus region of the brain, with voxel

*Table 14.* Dataset Information. The units for DISK-5k, DUTS, and Lucchi are images, while the unit for VOST is the number of videos, each containing several frames.

| Dataset Name | Training Set Size | Validation Set Size |
|---|---|---|
| DISK-5k | 3000 | 470 |
| DUTS | 10553 | 5019 |
| Lucchi | 165 | 165 |
| VOST | 619 | 24 |

resolutions of approximately 5x5x5nm.

For the video segmentation task, we selected the VOST (Tokmakov et al., 2023) dataset for fine-tuning. The VOST dataset is a collection of over 700 high-resolution videos focusing on complex object transformations. It is designed to evaluate video object segmentation methods under dynamic appearance changes, with dense instance mask labeling and a focus on spatiotemporal modeling.

Additionally, we select twelve other datasets to assess the model's generalization capability. The datasets used to test generalization performance on natural image segmentation include the following six: ADE20K (Xia et al., 2019), Cityscapes (Garcia-Garcia et al., 2017), COCO-stuff (Anwar et al., 2020), ECSSD (Tran et al., 2020), FSS (Li et al., 2020b), BIG (Cheng et al., 2020). ADE20K is a semantic segmentation dataset containing over 20,000 images annotated for 150 categories, including both "stuff" (e.g., sky, road) and "objects" (e.g., car, person). Cityscapes is a large-scale dataset of urban street scenes, with annotations for 30 classes across 5,000 finely labeled images. COCO-stuff extends the COCO dataset with pixel-level annotations for 172 categories, including both "things" and "stuff." ECSSD is a saliency dataset with 1,000 real-world images featuring complex textures and structures. FSS is a few-shot segmentation dataset with 1,000 classes that include many previously unseen or unannotated objects. BIG is a high-resolution semantic segmentation dataset with images ranging from 2048×1600 to 5000×3600, carefully labeled to align with PASCAL VOC 2012 standards. Among these, the first three datasets, which are significantly different from the fine-tuning dataset, are categorized as Group 1, while the latter three, which are more similar, are categorized as Group 2.

The datasets used to test segmentation on biomedical images include the following three: VNCIII (Gerhard et al., 2013), MitoEM-R (Wei et al., 2020), and MitoEM-H (Wei et al., 2020). VNCIII consists of a ground truth stack of 20 sections obtained using serial section Transmission Electron Microscopy (ssTEM) from the ventral nerve cord of the Drosophila melanogaster third instar larva. This dataset captures a volume approximately measuring 4.7 × 4.7 × 1 microns, with a pixel resolution of 4.6 × 4.6 nm and

section thickness ranging from 45 to 50 nm. It provides high-resolution insights into neural structures. MitoEM-R and MitoEM-H are 3D mitochondria segmentation datasets, each containing 1,000 consecutive slices. Both datasets include ground truth mitochondria instance labels for the first 500 slices, divided into training (slices 0–399) and validation (slices 400–499) subsets. To ensure high-quality annotations, every mitochondrion instance in the ground truth spans a minimum size of 2,000 voxels. While the annotations are comprehensive, refinement is encouraged, with contributors invited to report segmentation errors by specifying the (x, y, z) coordinates of erroneous regions.

The datasets used for video segmentation include the following three: UVO (Wang et al., 2021), VIPSeg (Miao et al., 2022), and PUMaVOS (Bekuzarov et al., 2023). UVO is a benchmark for open-world class-agnostic object segmentation in videos, offering significantly more videos and annotations than other datasets while presenting challenges such as crowded scenes and complex motions. VIPSeg is a large-scale dataset specifically designed for video panoptic segmentation tasks. PUMaVOS is a densely annotated dataset with 24 videos covering complex scenarios such as object parts, frequent occlusions, fast motion, and deformable objects, with an average video length of 29 seconds and a focus on benchmarking model performance.

All the above data sets are treated as instance segmentation.

## F. Metrics Details.

A total of four metrics are calculated in this article. Below we explain how to calculate them in detail:

First, we calculate the instance-level F1 score. Due to the characteristics of our SAM model, one prompt corresponds to the output of one target. However, in our datasets, only some data have one target per image, and most data have multiple instance-level targets per image. In order to adapt to the characteristics of SAM and a variety of datasets, we calculate the F1 score of each instance and then average it over all instances in the dataset:

$$F1 = \frac{2TP}{2TP + FN + FP}. \tag{18}$$

In this equation, TP refers to the number of correctly predicted pixels that overlap with the ground truth for a given instance. FP represents the pixels that are incorrectly predicted as belonging to the instance but do not overlap with the ground truth. FN are the pixels that belong to the ground truth instance but are missed by the model's prediction. Then there is the average intersection-over-union ratio, which is calculated in a similar way to F1:

$$mIou = \frac{TP}{TP + FP + FN}. \tag{19}$$

Next is the pixel-level Dice coefficient, which should be the same as F1, but the difference is that we take a pixel-weighted average of all instances on each image and then average all the images in the dataset.

Finally, we introduce the $\mathcal{J}\&\mathcal{F}$ metric, where $\mathcal{J}$ represents the IoU between the predicted mask and the ground truth, and $\mathcal{F}$ measures the alignment between the boundaries of the predicted mask and the ground truth boundaries. In essence, $\mathcal{F}$ is equivalent to the F1 score described earlier.

## G. Experiments Details.

In natural image segmentation, we use SAM (Kirillov et al., 2023) vit-h as the large model requiring fine-tuning and SAM vit-b as the guiding small model. Notably, the large model outperforms the small model in both zero-shot and standard fine-tuning settings. For the guiding small model, we follow the setup in (Ke et al., 2024), using the AdamW optimizer and an input size of 1024 × 1024. Similarly, we only fine-tune the mask decoder portion. The necessary guiding feature is extracted from the mask decoder. For the large model requiring fine-tuning, we use a similar setup to that of the small model, but with the addition of guiding feature based on the FRS module. Both the large model requiring fine-tuning and the small guiding model are trained for the same 20 epochs. We use a box as the prompt and evaluate only the first mask output.

The experimental setup for biomedical image segmentation uses the same hyperparameter settings as in natural image segmentation, and we sample the data into 165 slices of 768×1024 images. The key difference is that we employed a new model to provide the prompts and replaced HQ-SAM in the baseline with MedSAM, a SAM variant specifically designed for medical images.

For the video segmentation task, we use SAM2 (Ravi et al., 2024) hiera-large as the large model and hiera-tiny as the guiding small model. Similar to the previous settings, we use the AdamW optimizer and an input size of 1024×1024. In this task, we not only fine-tuned the mask decoder but also fine-tuned the memory encoder. For each iteration, a segment of the video is trained, sampling 16 consecutive frames, with prompts applied to the first frame. We use click prompts, where the first click is positioned at the center of the mask, and two additional clicks are randomly placed within the mask. When evaluating model performance, we use all available video frames but only provided prompts for the first frame. We compute the metrics for each instance in every frame and then average them across all frames. For SAM2, we employed three-point prompts: the first point is placed at the center of the mask, while the other two points are random within the mask.

## H. Extension to Other ViT-based Architectures

Although our main experiments are conducted on the original SAM for focus and clarity, the theoretical design of SHERPA is applicable to any ViT-based architecture that offers multiple model sizes (e.g., small and large variants). This includes models such as SAM variants, DINOv2, and MaskFormer.

To address this, we provide additional experiments on **HQ-SAM** and **Med-SAM**. Furthermore, to strengthen the evidence of SHERPA's generality and credibility, we also report results on **other architectures**, such as **DINOv2** and **MaskFormer**, where SHERPA can be directly applied.

|  | Valid | General |
|---|---|---|
| HQ-SAM (sft) | 0.9031 | 0.6931 |
| with SHERPA | 0.9082 | 0.7491 |
| MED-SAM (sft) | 0.8931 | 0.7952 |
| with SHERPA | 0.8945 | 0.8034 |
| DINOv2 (sft) | 0.9079 | 0.8031 |
| with SHERPA | 0.9094 | 0.8273 |
| MaskFormer (sft) | 0.8631 | 0.7242 |
| with SHERPA | 0.8753 | 0.7531 |

*Table 15.* Performance of SHERPA on additional ViT-based architectures.

These results confirm that SHERPA is broadly applicable beyond the original SAM family.

## I. Computational Overhead Analysis

Our method is designed specifically for fine-tuning, and the model's inference process remains identical to standard inference, thus incurring no additional overhead during inference. During training, the FRS module is lightweight and operates via simple projections, while the guiding SAM is significantly smaller than the large model, resulting in relatively low training costs.

To provide a quantitative comparison, we report the **training time** and **GPU memory consumption** for three settings: (1) fine-tuning the small model, (2) standard fine-tuning of the large model, and (3) fine-tuning the large model with our proposed method. All experiments were conducted on a single NVIDIA 3090 GPU.

## J. Fisher ratio Analysis

Averaging FR over a set of orthogonal directions thus quantifies **how much label-discriminative signal that sub-space carries**, independent of any classifier head.

**Interpretation**
The high Fisher ratio subspace contains **3×** more Fisher

|  | Train time | GPU mem |
|---|---|---|
| Fine-tuning small | 192 | 9.81 |
| SFT large model | 702 | 12.24 |
| SHERPA large model | 748 | 13.15 |

*Table 16.* Comparison of training time and GPU memory consumption for different fine-tuning strategies.

| Dataset | FR High | FR LoW | Gap |
|---|---|---|---|
| DISK-5K | **14.7** | 4.7 | ×3.1 |

*Table 17.* Fisher-ratio comparison between high FS and low FS subspaces on DISK-5K.

information than the low complement.

$\Rightarrow$ FRS has indeed concentrated almost all class-separable signals into the high block, while the low block is largely label-agnostic and thus a plausible carrier of generalization.

## K. Balancing Parameter $\lambda$.

The balancing parameter is crucial; a value that is too small can lead to insufficient feature focus, while a value that is too large may cause the large model to overly prioritize feature concentration, resulting in biased outputs. After testing various values in table 18, we found that a balancing parameter of 3 yields the best performance.

## L. Baselines Details.

We first introduce the baselines used in our article in Detail.

**Zero-shot.** For the simple Zero-shot baseline, we utilize a pre-trained model without any fine-tuning. The model is tested directly on the samples using prompts provided by the mask. The generalization performance observed in this setting is considered the upper bound for the generalization

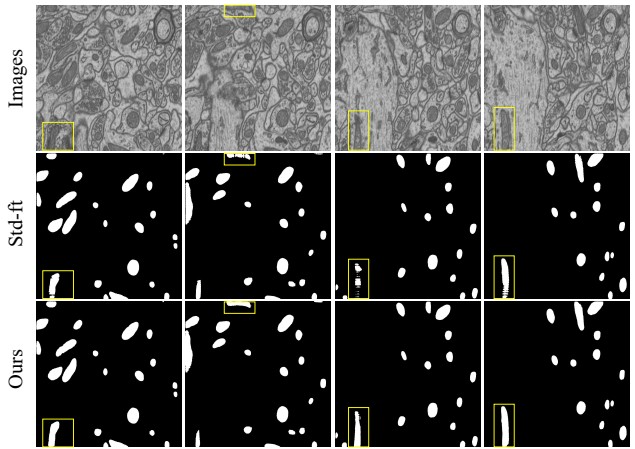

*Figure 5.* Visualization of fine-tuning results on Lucchi.

*Table 18.* Ablation study for Balancing Parameter $\lambda$ in DISK-5K. All metrics are instance-level F1 scores.

| $\lambda$ | 0.1 | 1 | 3 | 10 |
|---|---|---|---|---|
| Valid | 0.8701 | 0.8691 | **0.8855** | 0.8841 |
| General | 0.7895 | 0.7881 | **0.7930** | 0.7746 |

ability of the fine-tuned models.

**Std-ft.** For standard fine-tuning (Std-ft), we fine-tune only the `mask_decoder` of the SAM model, as well as the `mem_encoder` and `mask_decoder` of the SAM2 model. No additional operations are applied during the fine-tuning process.

$L^2$**-SP.** $L^2$-SP is a method proposed to mitigate generalization loss through a quadratic penalty. We applied quadratic penalties with different coefficients to the parts of the model requiring fine-tuning. Specifically, we tested four penalty coefficients ranging from $1e^{-2}$ to $1e^{-5}$, corresponding to $L^2$-SP2 to $L^2$-SP5, respectively. Among these, we selected $L^2$-SP3 and $L^2$-SP4, which demonstrated the best fine-tuning and generalization performance, to present the results in the main text.

**Ft-last.** This method aims to mitigate the loss of generalization ability by fine-tuning only the last few layers of the model. We tested fine-tuning the last 1 to 5 layers of the model, corresponding to Ft-last1 through Ft-last5. Ultimately, we selected Ft-last2 and Ft-last4, which exhibited the best performance, for presentation in the main text.

Next, we introduce several baselines that were not selected for inclusion in the main text due to their relatively poor performance.

**WiSE-ft.** WiSE-ft is a method originally designed to mitigate the generalization loss of CLIP models during fine-tuning. It achieves improved robustness by fusing model weights. We tested various parameter combinations and concluded that this method is not suitable for image segmentation tasks.

**Stochastic-ft.** In this approach, we randomly fine-tune a certain percentage of the blocks in the pre-trained network. Despite testing multiple configurations, we found that this method is also not effective for image segmentation tasks.

## M. Parameter sensitivity

SHERPA is robust to subspace estimation parameters. Performance is **stable** for $m > 50$ and $k$ between 2–5 (Fisher ratio coverage 50%–80%). We recommend $m = 50$ and $k = 3$ (75% coverage) as default.

Additionally, we will provide further performance curves for $(m, k)$ in the 6.

**Effect of** $m$ With $m = 50$, the subspace is well estimated; further increases yield only marginal gains. Smaller $m$ may

*Table 19.* Performance with different $m$ values.

| $m$ | Valid | Generalization |
|---|---|---|
| 10 | 0.8754 | 0.7631 |
| 20 | 0.8824 | 0.7757 |
| 50 | 0.8855 | 0.7930 |
| 100 | 0.8842 | 0.7935 |
| 200 | 0.8857 | 0.7926 |

cause unreliable estimates.

**Effect of $k$** We do not select $k$ directly, but use the proportion of Fisher ratio for task-relevant channels as the criterion. Best performance is at 75% coverage ($k = 3$ for SAM). Too small $k$ lacks task-relevant features; too large $k$ introduces noise and degrades both fine-tuning and generalization.

In the table below, we also provide the direct relationship between the value of $k$ and the model performance (note that there are **differences** from Table 5 due to the rounding of Fisher ratio quartiles).

*Table 20.* Performance with different $k$ values and Fisher ratio coverage.

| $k$ | Ratio | Valid | Generalization |
|---|---|---|---|
| 0 | 0.00 | 0.8789 | 0.7234 |
| 1 | 0.34 | 0.8816 | 0.7531 |
| 2 | 0.57 | 0.8850 | 0.7891 |
| 3 | 0.76 | **0.8855** | **0.7930** |
| 4 | 0.84 | 0.8842 | 0.7735 |
| 5 | 0.87 | 0.8738 | 0.7758 |
| 56 | 1.00 | 0.8695 | 0.5257 |

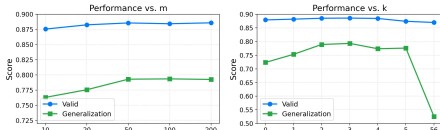

*Figure 6.* Performance curves for different values of $m$ and $k$. SHERPA is robust for $m > 50$ and $k$ between 2–5.

## N. Robustness Test

We test several scenarios involving noisy or imperfect guidance, such as underfitting caused by fewer training epochs, and bias caused by incorrect labels. In all these experiments, the large model remains unchanged.

We observe that even in these extreme conditions, the generalization ability of the large model is not compromised. This is due to the inherent robustness of our method by design, as explained in the following section.

At the same time, we note that while guidance from a small model trained with incorrect labels can affect the large

*Table 21.* Robustness test under different guidance settings.

| Setting | Valid | Generalization |
|---|---|---|
| zero-shot | 0.6570 | 0.8483 |
| std-ft | 0.8728 | 0.7236 |
| 2 epoch | 0.8731 | 0.7623 |
| 5 epoch | 0.8745 | 0.7731 |
| 10% label noisy | 0.8804 | 0.7843 |
| 20% label noisy | 0.8736 | 0.7626 |
| 50% label noisy | 0.8492 | 0.7572 |
| full (20 epoch + no noisy) | 0.8855 | 0.7930 |

model's validation performance to some extent, it does not fall below that of standard fine-tuning unless the label noise is extremely severe (i.e., more than 50% of the labels are noisy).

## O. Additional Baselines Comparison

To comprehensively evaluate SHERPA, we provide additional comparisons with representative methods from three related fields: Parameter-Efficient Fine-Tuning (PEFT), Knowledge Distillation (KD), and standard Regularization methods. All experiments in this section are evaluated on the DISK-5K dataset.

### O.1. Comparison with PEFT Methods

While PEFT methods focus on updating a minimal number of parameters to maintain efficiency, SHERPA focuses on how to guide the update process to preserve pretrained generalization. As shown in Table 22, although PEFT methods like AdaptFormer and Gradient-based parameter selection can achieve strong validation performance, they still suffer from significant generalization degradation. SHERPA better preserves generalization while remaining competitive on the target task, making it complementary to PEFT techniques.

*Table 22.* Comparison with representative PEFT methods on DISK-5K. All metrics are F1 scores.

| Method | Valid | Generalization |
|---|---|---|
| Std-ft | 0.8728 | 0.7236 |
| VPT | 0.8814 | 0.6723 |
| AdaptFormer | 0.8931 | 0.7102 |
| Grad.-based param. selection | 0.9031 | 0.6924 |
| SHERPA (Ours) | **0.8855** | **0.7930** |

### O.2. Comparison with Knowledge Distillation Methods

Although SHERPA shares a similar training form with Knowledge Distillation (KD), it fundamentally differs in objective and mechanism. KD transfers predictive behavior by matching logits or full features, whereas SHERPA uses the small model solely to identify a task-relevant subspace

constraint. Table 23 demonstrates that SHERPA significantly outperforms standard KD and recent KD variants in preserving generalization.

*Table 23.* Comparison with Knowledge Distillation (KD) methods on DISK-5K.

| Method | Valid | Generalization |
|---|---|---|
| std-KD | 0.8793 | 0.7351 |
| FEED | 0.8801 | 0.7342 |
| DLIM-Det | 0.8698 | 0.7262 |
| SHERPA (Ours) | **0.8855** | **0.7930** |

### O.3. Comparison with Regularization Methods

Standard regularization methods typically restrict deviation from pretrained parameters uniformly, which may suppress both harmful and beneficial updates. SHERPA, in contrast, applies selective guidance within the high-Fisher-ratio subspace. As shown in Table 24, standard regularization methods (such as EWC and SI) preserve more generalization than standard fine-tuning but often at the cost of weaker task adaptation. SHERPA achieves a strictly better trade-off.

*Table 24.* Comparison with standard Regularization methods on DISK-5K.

| Method | Valid | Generalization |
|---|---|---|
| $L^2$-SP4 | 0.7782 | 0.7025 |
| EWC | 0.8415 | 0.7682 |
| SI | 0.8341 | 0.7831 |
| SHERPA (Ours) | **0.8855** | **0.7930** |

## P. Additional Visualization Results

Figure 7 shows additional segmentation results on natural images from the DIS5K dataset. Figure 5 presents results on biomedical images from the Lucchi dataset, and Figure 8 shows video segmentation results on the VOST dataset.

## Q. Visualization of Feature.

We visualized the Feature of each channel after applying Std-ft and our method, as shown in Figure 9.

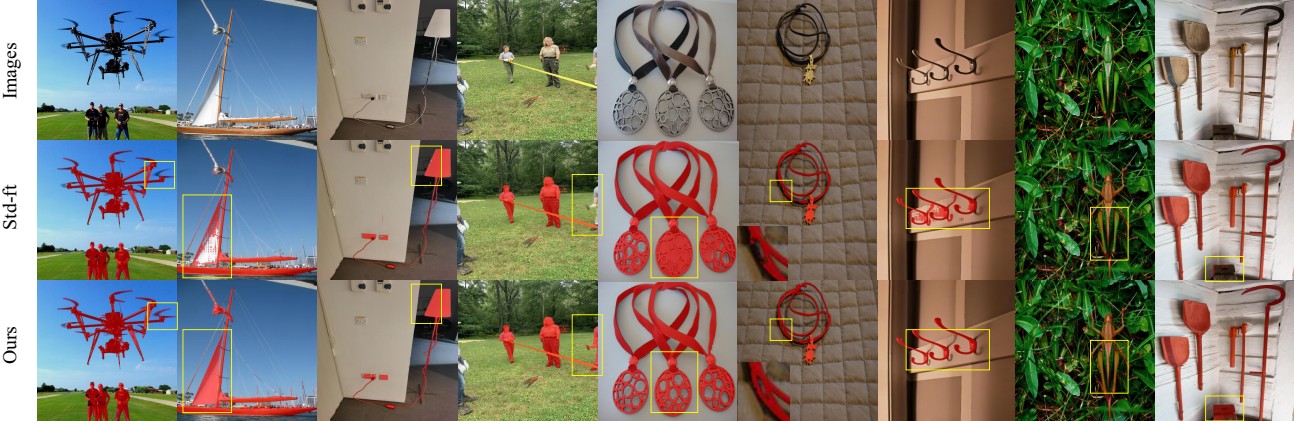

*Figure 7.* Visualization of fine-tuning results on DISK-5K.

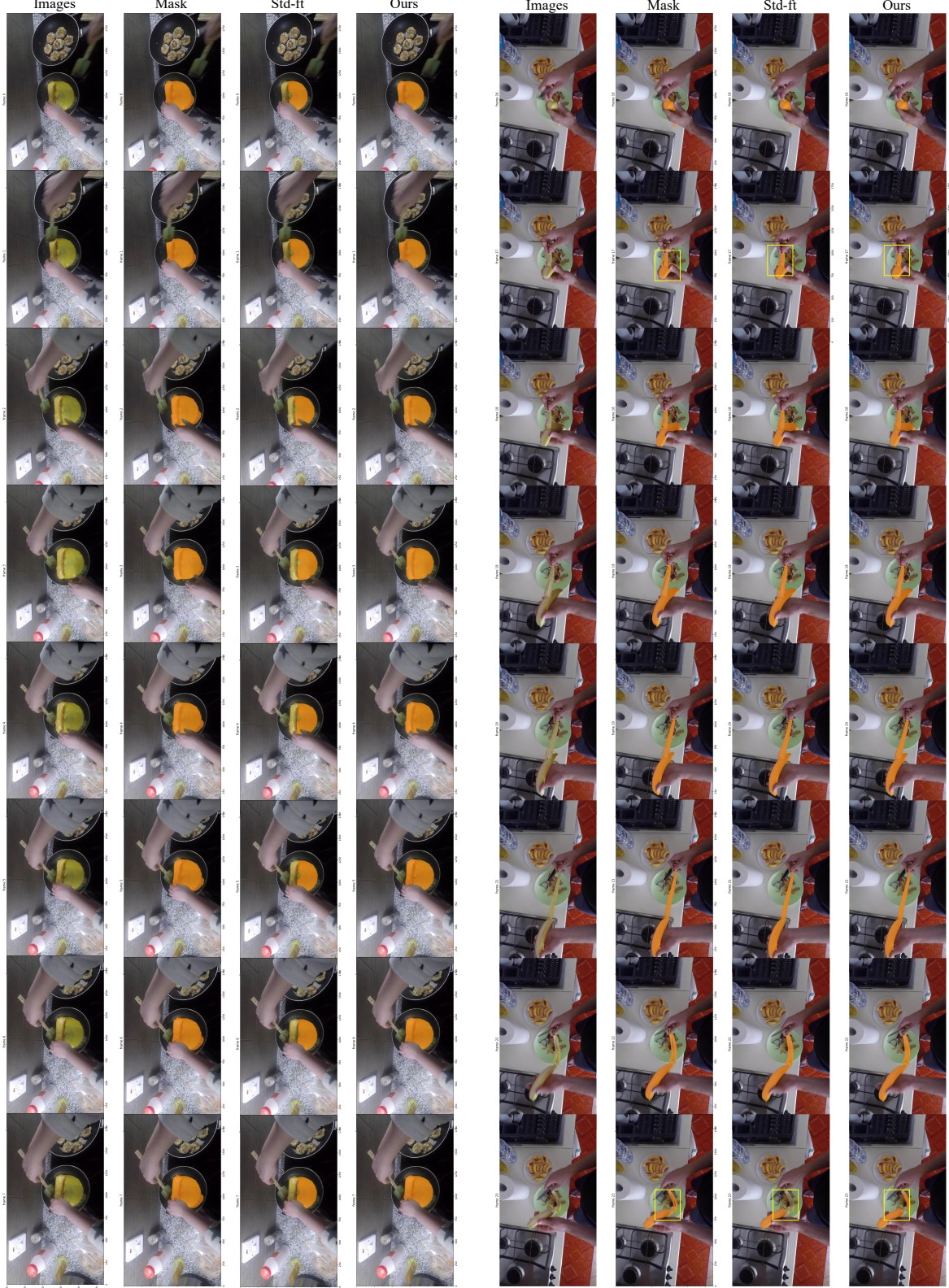

*Figure 8.* Visualization of the video segmentation results.

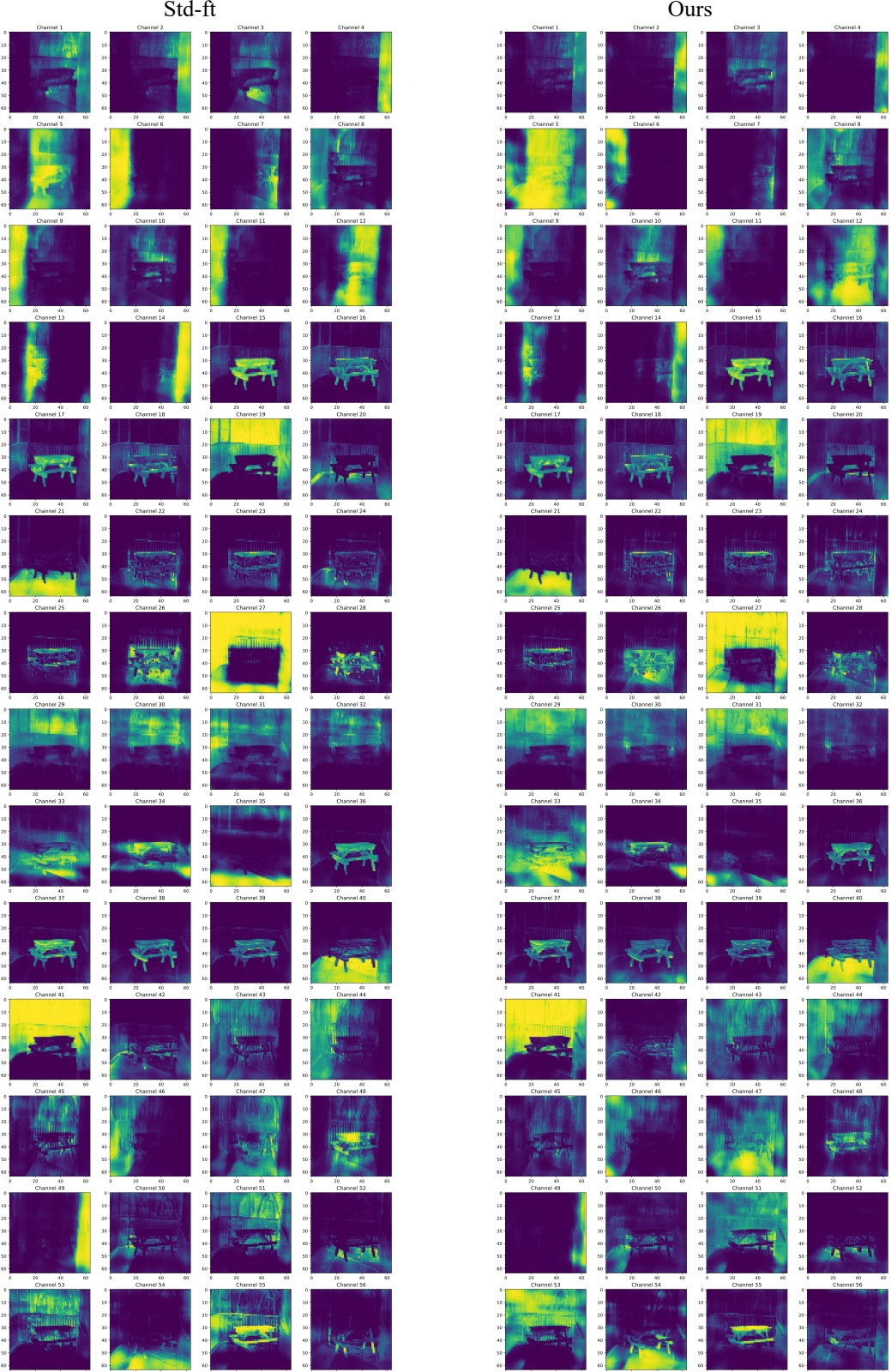

*Figure 9.* Visualization of the detailed Feature of each channel.

