# OpenReview forum: "SHERPA: Fine-tuning Segment Anything Models with Task-relevant Guidance"
_ICML.cc/2026/Conference — ICML 2026 regular_

### Official Review · Reviewer_ttRZ · 2026-03-12

**Soundness:** 2
**Presentation:** 3
**Significance:** 2
**Originality:** 2
**Overall Recommendation:** 3
**Confidence:** 3

**Summary:**

This paper introduces a framework for fine-tuning large Segment Anything Models (SAMs) while attempting to preserve their generalization ability. The authors argue that standard fine-tuning on task-specific data often causes large SAM models to overfit and lose the broad generalization obtained during pretraining. To address this issue, the paper proposes leveraging smaller SAM models trained on specific tasks to guide the fine-tuning of a larger SAM.

The method consists of two main components. First, the Fisher Ratio Separation module identifies and separates task-relevant features from the large model’s representation space, constructing a subspace intended to preserve useful information while avoiding excessive compression during fine-tuning. Second, the Guiding Feature Extraction module extracts representative task-relevant features from a smaller SAM model and uses them to guide the fine-tuning process of the larger model. The idea is that smaller models capture more focused task-relevant features, which can be transferred to the larger model to improve specialization without harming generalization.

**Compliance With Llm Reviewing Policy:**

Affirmed.

**Final Justification:**

While the rebuttal addressed the technical implementation details, it did not resolve the fundamental question of whether this is a practical, scalable solution that the broader community would adopt. Therefore, I maintain my original score as weak reject.

**Key Questions For Authors:**

1. In which scenarios would practitioners prefer the proposed framework over simpler fine-tuning or distillation methods, given the additional training stages and computational overhead?
2. Why the Fisher ratio is the most appropriate criterion for separating task-relevant features?
3. Why guidance from smaller models is preferable to alternative regularization or representation preservation methods?

**Limitations:**

No. The paper could more clearly discuss practical limitations.
Explicit discussion of these aspects would improve transparency.

**Strengths And Weaknesses:**

## Strengths
* The proposed framework is reasonably well structured and includes two clearly defined components (FRS and GFE).
* The authors provide a decent number of empirical evaluations and ablation studies to analyze the effect of different modules.
* A unique approach towards the segmentation problem, but the author could have provided additional evidence (explained in the weakness).
* The overall writing and presentation style is good.

## Weaknesses

* While the method is technically reasonable, the empirical gains appear relatively small compared to the baseline fine-tuning approaches.
* The proposed framework introduces additional computational overhead, since it requires training and maintaining an auxiliary smaller SAM model and performing feature extraction and guidance during fine-tuning.
* The paper does not clearly demonstrate that the added complexity is justified by the observed improvements.
* The problem addressed is relevant; however, the practical impact of the proposed solution is unclear, considering the computational overhead.
* The improvements over standard fine-tuning are relatively modest. It remains unclear whether practitioners would adopt this approach given the added complexity.
* The paper studies SAM fine-tuning strategies but does not discuss prior empirical analyses, such as How to build the best medical image segmentation algorithm using foundation models (arXiv:2404.09957), which already evaluates multiple SAM fine-tuning approaches across datasets.
* Lack of detailed comparison vs LoRA/Peft model: The author compares LoRA-based models with and without SHEPRA. for Disk-5k

### Originality
* The combination of Fisher ratio separation and feature guidance is somewhat novel, but overall, the method appears to be a combination of existing ideas rather than a fundamentally new approach.

---

> ### Author Rebuttal · Authors · 2026-03-31
>
> # Q1 / W1–W5
>
> We thank the reviewer for raising SHERPA’s practical relevance. At core, W1–W5 and Q1 ask whether its added training complexity is worthwhile and when it is preferable. We agree the current paper does not explain this clearly and will clarify its scope in revision.
>
> ## What SHERPA improves
>
> SHERPA is not intended to replace standard fine-tuning or PEFT/KD as a universal recipe. We also agree that, in some settings, its task-specific gain is modest.
>
> However, for a large SAM, pretrained generalization is not a secondary property but a core part of its value. If fine-tuning degrades it, the model may gain specialization at the cost of SAM’s broad utility.
>
> SHERPA therefore focuses on generalization retention, improving post-adaptation retention by +8.1 / +8.4 / +12.9 / +11.1 on DISK-5K / DUTS / Lucchi / VOST, respectively.
>
> ## When practitioners would prefer SHERPA
>
> When the deployment goal is not only to adapt to the target domain, but also to preserve strong generalization ability after adaptation, SHERPA is a more suitable choice than standard FT/KD.
>
> SHERPA is suitable when the deployment distribution is not fixed and a single model is expected to cover both the specialized task and the original generalization ability.
>
> A practical example is an **interactive annotation tool**, where a team may adapt SAM using a small amount of in-house data to better match its annotation style, while still requiring the model to remain effective on other objects and scenes.
>
> ## Computational cost
>
> If the goal is to preserve generalization ability after adaptation, the comparison should not be limited to standard fine-tuning, but should include other retention-oriented solutions, such as maintaining both the original and specialized models, or re-adapting after forgetting when out-of-domain needs arise.
>
> These alternatives may not increase single-run training cost, but they add deployment and maintenance burden. In contrast, SHERPA shifts the extra cost to training while keeping single-model deployment; a lower-cost variant in **C.4 / Table 13** further reduces the extra training cost to about **9%**.
>
>
>
> # W6 / W7
>
> Thank you for pointing out this relevant benchmark, which we will discuss in revision. It is complementary to our work: it compares SAM adaptation recipes mainly by in-domain performance, while our focus is on post-adaptation generalization retention.
>
> Beyond the existing DISK-5K experiment, we further evaluated SHERPA with LoRA and Adapter on **DUTS / Lucchi**. Below we report **Gen.**:
>
> - **LoRA**: **0.7793 / 0.6815**
> - **LoRA + SHERPA**: **0.8183 / 0.7721**
> - **Adapter**: **0.7285 / 0.6734**
> - **Adapter + SHERPA**: **0.8294 / 0.7785**
>
> These results show that SHERPA complements PEFT: it improves generalization with comparable or better validation performance.
>
> # Q2
>
> To clarify, we do not claim that Fisher ratio is the most appropriate criterion among all possible choices; rather, for our objective, it is a principled and effective criterion.
>
> Our goal is to identify task-discriminative directions, rather than generally “important” or high-response features. Fisher ratio is a supervised criterion directly aligned with this goal, as it measures label separability through between-class separation and within-class compactness. It also naturally yields a stable and operational subspace decomposition, which is important for applying guidance only to the task-relevant part of the representation.
>
> We further support this choice with two controls: random subspace and shuffled Fisher labels. Below we report **Valid / Gen.** on **DISK-5K / DUTS / Lucchi**:
>
> - **SHERPA**: **0.8855 / 0.7930**, **0.9495 / 0.8284**, **0.9231 / 0.7856**
> - **Random subspace**: **0.8632 / 0.6932**, **0.9341 / 0.7152**, **0.8942 / 0.6521**
> - **Shuffled Fisher**: **0.8231 / 0.6515**, **0.9189 / 0.7626**, **0.9051 / 0.6355**
> - **Std-ft**: **0.8728 / 0.7236**, **0.9402 / 0.7575**, **0.9113 / 0.6750**
>
> Taken together, these results support Fisher ratio as an appropriate criterion for SHERPA: the benefit comes specifically from selecting a label-aware discriminative subspace with strong between-class separation and low within-class variation, rather than from restricting optimization to an arbitrary subspace of the same dimension.
>
> # Q3
>
> Compared with regularization or representation preservation methods, which mainly constrain how far the model moves from its pretrained representation, SHERPA uses guiding features from a smaller model to provide an explicit task-oriented signal in the task-relevant subspace while preserving the broad capabilities of the large model.
>
> Because of stronger capacity constraints, smaller models are more likely to capture **compact, task-sufficient cues**, making them suitable guidance sources.
>
> Table 1 already includes KL-SP, L2-SP, and InfoSAM; we further tested EWC and SI. On DISK-5K, L2-SP4, EWC, and SI achieve validation/generalization scores of **0.7782/0.7025, 0.8415/0.7682, and 0.8341/0.7831**.

---

> > ### Author Rebuttal · Reviewer_ttRZ · 2026-04-04
> >
> > I thank the authors for the detailed rebuttal and the additional experiments provided. The new results comparing SHERPA against control subspaces (Random and Shuffled Fisher) and other regularization methods (EWC, SI) are technically informative and clarify the effectiveness of the Fisher ratio criterion. The additional results make the results more stronger.
> >
> > However, despite these technical clarifications, the paper lacks concrete motivation and practical positioning. While the authors demonstrate that SHERPA is effective for 'generalization retention', the added training complexity and the niche use case (a single model required to be both specialized and generalized) do not feel sufficiently justified as a major advancement for a general ML audience. The task-specific gains are still modest in several settings. Consequently, while the rebuttal addressed my technical questions, it did not fundamentally change my assessment of the paper's significance. I will maintain my current score.

---

> > > ### Author Response · Authors · 2026-04-05
> > >
> > > Thank you for the follow-up. We are glad that the additional experiments and clarifications “addressed the technical questions” and “make the results stronger”. The only remaining concern is the assessment of the paper’s significance. However, this could be a **misunderstanding** on our real target in the paper.
> > >
> > > *“The paper lacks concrete motivation and practical positioning.”*
> > >
> > > (1) For segmentation foundation models, broad zero-shot/general capability is a **core part** of the model’s value. In practice, an adapted model is often still expected to remain useful beyond the task-specific distribution, including under broader prompts, scenes, categories, or mild downstream shift. When adaptation substantially degrades this capability, it undermines a major benefit of pretraining and reduces the usefulness of the adapted model in deployment. 'Generalization retention' is thus an important objective for adapting pretrained segmentation models, and **there are plenty of seminal works along this direction** [1,2,3].
> > >
> > > (2) The paper studies this objective directly: adapting a pretrained segmentation foundation model to a specific target task while preserving its pretrained general utility, rather than optimizing only task-specific accuracy. Under this objective, SHERPA improves post-adaptation generalization retention by +8.1 / +8.4 / +12.9 / +11.1 on DISK-5K / DUTS / Lucchi / VOST, while maintaining competitive target-task performance. Noticeably, we do not see *“the task-specific gains are still modest in several settings”* as a weakness, since **this is not our main objective**.
> > >
> > > (3) Last but not least, the paper’s significance has been well recognized by other reviewers.
> > >
> > >
> > >
> > > In summary, since the reviewer acknowledged that “the technical questions have been addressed”, and “the additional results make the paper stronger”, the only remaining concern on “the paper’s significance” tends to be subjective. We have provided our last defense, and we hope it could be taken into consideration during the final decision.
> > >
> > >
> > >
> > > [1] Robust fine-tuning of zero-shot models (CVPR 2022)
> > >
> > > [2] MERGETUNE: Continued Fine-Tuning of Vision-Language Models (ICLR 2026)
> > >
> > > [3] Robust Fine-tuning of Vision-Language-Action Robot Policies via Parameter Merging (ICLR 2026)

---

### Official Review · Reviewer_9buR · 2026-03-12

**Soundness:** 3
**Presentation:** 3
**Significance:** 3
**Originality:** 3
**Overall Recommendation:** 4
**Confidence:** 3

**Summary:**

The paper proposes SHERPA, a fine-tuning method for SAM. The goal is to adapt SAM to a specific task while preserving its zero-shot ability as much as possible. To do this, the authors use a smaller SAM to learn task-relevant features and guide a larger SAM during fine-tuning. They use the Fisher Ratio to separate task-specific features, and Guiding Feature Extraction to obtain the relevant guidance. The method performs well in keeping a good balance between finetuning and generalization.

**Compliance With Llm Reviewing Policy:**

Affirmed.

**Final Justification:**

The authors addressed the most of my concerns in the rebuttal. I keep my score as Weak Accept.

**Key Questions For Authors:**

Is learning from a small model acting mainly as a stronger form of regularization, since larger models tend to overfit more easily? Could the paper include some experiments to show that the gain is not simply from regularization, for example, by comparing with standard regularization methods?

For other questions, please see the weakness section.

**Limitations:**

The paper would benefit more from a limitations section about some limitations of SHERPA and future directions.

**Strengths And Weaknesses:**

Strengths:
1. The idea is interesting and relevant. Fine-tuning a foundation model while keeping its zero-shot performance is an important problem, and the proposed design is effective for this goal.
2. The writing is clear and easy to follow.
3. There are a lot of experiments and a large set of evaluation benchmarks, which helps show the strong performance of the model.

Weaknesses:
1. There is some risk of propagating bias or errors from the small model. If the small model contains artifacts or wrong priors, it seems hard for the large model to correct them in the current setting.
2. There is additional training overhead. Since the method involves training with two models, showing some FLOPs or training cost comparison would strengthen the paper and help confirm that using both models is worthwhile.
3. More feature visualization would strengthen the intuition and improve the understanding of how the module works, especially for the task-relevant features.

---

> ### Author Rebuttal · Authors · 2026-03-31
>
> # Q1
>
> We thank the reviewer for this key question. We do **not** consider SHERPA to be a regularization method, although any mechanism that restricts harmful updates may naturally exhibit some regularizing effect in practice. To directly address this point, our main results table (**Table 1**) already includes two representative regularization baselines (**KL-SP** and **L2-SP**), and we further provide additional comparisons with standard regularization methods.
>
> |        | DISK-5K Valid | Gen.   |
> | ------ | ------------- | ------ |
> | L2-SP4 | 0.7782        | 0.7025 |
> | EWC    | 0.8415        | 0.7682 |
> | SI     | 0.8341        | 0.7831 |
> | SHERPA | 0.8855        | 0.7930 |
>
> Empirically, if SHERPA’s gain mainly arose from stronger regularization, these methods should achieve a task/generalization trade-off comparable to SHERPA. However, they all perform clearly worse than SHERPA. This indicates that SHERPA’s improvement cannot be simply attributed to generic regularization. More importantly, standard regularization usually preserves more generalization only at the cost of weaker task adaptation, whereas SHERPA better retains generalization **without sacrificing, and often while maintaining stronger, task adaptation performance**. This is qualitatively different from being “just a stronger regularizer.”
>
> From a methodological perspective, standard regularization methods constrain deviation from the pretrained parameters or representations in a **global or uniform** manner. They do not distinguish between directions that are truly needed for the current task and directions that support the model’s broader generalization ability, and thus are closer to suppressing updates overall. In contrast, the key idea of SHERPA is that the small model provides a cleaner task-relevant structure, FRS further identifies a high-Fisher-ratio subspace, and the large model is guided mainly within this subspace. In other words, SHERPA is not simply about “updating less,” but about **updating in more appropriate directions**.
>
> # W1
>
> Thank you for this question. In SHERPA, the small model is not used as a prediction target or hard teacher. Instead, it provides guidance in the Fisher-selected task-relevant subspace, rather than imposing its full representation or logits on the large model. This means that artifacts or biased priors in the guiding model are not transferred wholesale, while the ground-truth task supervision still allows the large model to correct moderate mismatched guidance.
>
> We further evaluate this in **Appendix N, Table 21**. Under imperfect guidance (2/5 training epochs or 10%/20% label noise), SHERPA still achieves better generalization than standard fine-tuning (**0.7623 / 0.7731 / 0.7843 / 0.7626 vs. 0.7236**), which supports its robustness to moderate guide imperfections.
>
> # W2
>
> We thank the reviewer for raising this practical concern. In our setting, standard fine-tuning requires **0.982 EFLOPs**, while SHERPA requires **1.163 EFLOPs**.
>
> Importantly, this overhead is incurred **only during training**: at inference time, SHERPA uses **only the large model**, and therefore introduces **no extra test-time FLOPs, latency, or deployment cost**.
>
> We believe this trade-off is worthwhile when the goal is not only target-task adaptation, but also **retaining the pretrained model’s generalization ability after adaptation**. In our experiments, SHERPA consistently improves generalization retention over standard fine-tuning (e.g., **+8.1 / +8.4 / +12.9 / +11.1** on **DISK-5K / DUTS / Lucchi / VOST**). Thus, the extra training cost is used to preserve broad capability in a single deployed model, rather than to increase inference-time model size or complexity. We also note that a lower-cost variant in **C.4 / Table 13** further reduces the extra training overhead to about **9%(GPUhS)**.
>
> Due to space, we keep the response brief here. If helpful, a fuller discussion is provided in **ttRZ (Q1/W1–W5)**, and we will also clarify this point in the revision.
>
> # W3
>
> We agree that additional feature visualizations would further strengthen the intuition and make the mechanism clearer, especially for the task-relevant features. The intended effect of SHERPA is that the small model provides a cleaner task-relevant structure, while FRS identifies high-Fisher-ratio directions and guides the large model mainly within this subspace rather than uniformly over the full feature space. This is also consistent with our component-wise results, where removing these components weakens the task/generalization trade-off. Due to rebuttal format, we cannot include a full set of additional visualizations here, but we will add them in the revision to make this intuition more explicit.

---

> > ### Author Rebuttal · Reviewer_9buR · 2026-04-03
> >
> > Thanks to the authors for the rebuttal. The additional results will make the paper stronger. I will keep my initial score.

---

> > > ### Author Response · Authors · 2026-04-04
> > >
> > > Thank you for your acknowledgement. We are very glad to know that you found our rebuttal fully addressed your concerns. We also sincerely appreciate your comment that the additional results will strengthen the paper. We will incorporate these clarifications and results carefully in the revision.

---

### Official Review · Reviewer_D17h · 2026-03-13

**Soundness:** 3
**Presentation:** 3
**Significance:** 3
**Originality:** 3
**Overall Recommendation:** 4
**Confidence:** 4

**Summary:**

The paper presents SHERPA, a fine-tuning framework designed to improve the generalization ability of Segment Anything Models (SAMs) while maintaining strong performance on specific tasks. SHERPA achieves this by using a smaller SAM model to guide the fine-tuning of a larger SAM model through task-relevant features. The method includes two key components: the Fisher Ratio Separation (FRS) module, and the Guiding Feature Extraction (GFE) module. The paper demonstrates SHERPA's effectiveness across various datasets, showing significant improvements in task performance and better retention of generalization ability compared to standard fine-tuning approaches.

**Compliance With Llm Reviewing Policy:**

Affirmed.

**Final Justification:**

The author's rebuttal has improved the paper and addressed my main concerns. It positively changed my evaluation, and I will raise my score accordingly.

**Key Questions For Authors:**

See above.

**Limitations:**

Yes.

**Strengths And Weaknesses:**

### Strengths
- The proposed method is backed by a theoretical framework, which effectively address the challenges of retaining generalization during fine-tuning.
- The paper is well-structured, with clear explanations of the methodology, experiments, and results.
- SHERPA combines existing ideas in a novel way, showing originality in its methodology and application.


### Weaknesses
- While LoRA and adapters are referenced, they are relatively older methods. The paper misses comparisons with more recent works in the visual PEFT field, as well as pretraining methods. Given the paper's focus on visual fine-tuning, comparative experiments with these methods would have been valuable.
- The method design in Sec 3.5 is similar to knowledge distillation. The authors should emphasize the relevance and difference of this method to knowledge distillation-based techniques and highlight SHERPA's innovative aspects in comparison.
- It is unclear how robust the method is in more complex or atypical domains. This paper provides limited empirical analysis on how sensitive the method is to subspace estimation parameters $(m, k)$.
- The introduction of the Fisher ratio in Sec 3.1 seems a little bit abrupt. Why the Fisher ratio is suitable for this context before detailing its use?

---

> ### Author Rebuttal · Authors · 2026-03-31
>
> # W1
>
> We agree that adding visual PEFT baselines strengthens the evaluation. We therefore include representative PEFT methods here; the submission already compares with recent fine-tuning baselines such as FisherTune and InfoSAM (Table 1).
>
> | Method                       |  Valid |   Gen. |
> | ---------------------------- | -----: | -----: |
> | Std-ft                       | 0.8728 | 0.7236 |
> | VPT                          | 0.8814 | 0.6723 |
> | AdaptFormer                  | 0.8931 | 0.7102 |
> | Grad.-based param. selection | 0.9031 | 0.6924 |
> | **SHERPA**                   | 0.8855 | 0.7930 |
>
> These results suggest that PEFT and SHERPA address different questions: PEFT studies which parameters to update, while SHERPA studies how to guide adaptation to preserve pretrained generalization. SHERPA is therefore complementary to PEFT and can be combined with LoRA/adapters. Unlike pretraining methods, SHERPA targets downstream fine-tuning. Its main advantage is post-adaptation generalization retention with competitive task performance.
>
> # W2
>
> Although SHERPA may resemble KD in training form, it differs from standard KD in objective, mechanism, and model role.
> (1) Objective: KD transfers predictive behavior/representations from teacher to student; SHERPA aims to preserve the large SAM’s original generalization while adapting it to the target task.
> (2) Mechanism: SHERPA does not match logits, predictions, or full features. Instead, it uses the small SAM only to identify a Fisher-selected task-relevant subspace, and guides the large SAM only in that subspace.
> (3) Model role: in KD, the teacher is the target to imitate; in SHERPA, the small model is only a guide for extracting task relevance.
>
> We also compared with several KD-style baselines on DISK-5K:
>
> | Method     |  Valid |   Gen. |
> | ---------- | -----: | -----: |
> | std-KD     | 0.8793 | 0.7351 |
> | FEED       | 0.8801 | 0.7342 |
> | DLIM-Det   | 0.8698 | 0.7262 |
> | **SHERPA** | 0.8855 | 0.7930 |
>
> SHERPA is distinct from KD and better preserves generalization.
>
> # W3
>
> We agree that robustness in atypical settings and sensitivity deserve clearer discussion. Existing evidence already suggests SHERPA is robust: Table 6 shows that it does not rely on a single task-relevant channel ratio; Appendix M reports parameter sensitivity; Appendix N reports robustness under imperfect guidance; and Table 5 shows that the benefit of FRS is consistent across datasets and settings.
>
> To address the reviewer’s concern more directly, we further evaluate SHERPA in several more atypical settings. Below we report **Std-ft -> SHERPA** in **Valid / Gen.**:
>
> - **Point prompts** on DISK-5K: **0.8623 / 0.7102 -> 0.8734 / 0.7812**
> - **Coarse mask prompts**: **0.8712 / 0.7467 -> 0.8789 / 0.7821**
> - **Background segmentation**: **0.6654 / 0.7102 -> 0.6874 / 0.7832**
> - **Partial segmentation**: **0.8293 / 0.7421 -> 0.8392 / 0.8412**
>
> SHERPA also remains robust under imperfect guidance. With guiding-model label noise of **\(10\%\)**, **\(20\%\)**, and **\(50\%\)**, SHERPA achieves **0.8804 / 0.7843**, **0.8736 / 0.7626**, and **0.8492 / 0.7572**, respectively, compared with **0.8728 / 0.7236** for Std-ft.
>
> Overall, these results indicate that SHERPA is not brittle: its gains remain consistent across atypical prompting/segmentation settings and persist even when the guiding model is imperfect.
>
> # W4
>
> We agree that the motivation for introducing Fisher ratio in Sec. 3.1 is not sufficiently explained, and we will clarify this in the revision.
>
> Fisher ratio is suitable here because SHERPA seeks task-discriminative directions in the pretrained representation, rather than features that are merely high-response or generally important. As a supervised criterion, Fisher ratio explicitly favors directions with strong between-class separation and small within-class variation, which aligns with our goal of isolating the task-relevant subspace. It also yields a practical decomposition into task-relevant and complementary subspaces, which is important because SHERPA only injects guidance into the former.
>
> We further support this choice with two controls: random subspace and shuffled Fisher labels. Below we report **Valid / Gen.** on **DISK-5K / DUTS / Lucchi**:
>
> - **SHERPA**: **0.8855 / 0.7930**, **0.9495 / 0.8284**, **0.9231 / 0.7856**
> - **Random subspace**: **0.8632 / 0.6932**, **0.9341 / 0.7152**, **0.8942 / 0.6521**
> - **Shuffled Fisher**: **0.8231 / 0.6515**, **0.9189 / 0.7626**, **0.9051 / 0.6355**
> - **Std-ft**: **0.8728 / 0.7236**, **0.9402 / 0.7575**, **0.9113 / 0.6750**
>
> Taken together, these results support Fisher ratio as an appropriate criterion for SHERPA: the benefit comes specifically from selecting a label-aware discriminative subspace with strong between-class separation and low within-class variation, rather than from restricting optimization to an arbitrary subspace of the same dimension.

---

> > ### Author Rebuttal · Reviewer_D17h · 2026-04-04
> >
> > Thank you for the rebuttal and the additional experiments. The new results improve the paper. I will keep the score.

---

> > > ### Author Response · Authors · 2026-04-04
> > >
> > > We are glad that your previous concerns have been **fully addressed** by our responses. Given that no more concerns are raised, we kindly ask you to consider adjusting the initial score (which was negative) accordingly.

---

### Decision · Program_Chairs · 2026-04-30

**Decision:**

Accept (regular)

**Comment:**

This paper presents a new finetuning strategy for segment anything models (SAMs), which leverages a smaller SAM to guide the finetuning of larger SAMs to improve performance on specific tasks while maximizing the retention of the generalization capability of original SAMs. The reviewers recognized the interesting and timely topic,  theoretical foundation for the proposed method, and extensive experiments. They also unanimously appreciated the clarity of the manuscript. However, they also raised multiple critical concerns with
1. additional training overhead (9buR, ttRZ)
2. lack of comparisons with parameter-efficient finetuning (D17h, ttRZ)
3. unclear advantages over knowledge distillation (D17h)
4. potential vulnerability due to bias or errors from the small model (9buR)
5. limited empirical analysis on complex and atypical domains (D17h)
6. limited comparisons with standard regularization methods (9buR)
7. limited performance gain compared to finetuning baselines (ttRZ), and
8. incremental novelty (ttRZ).

The authors responded to these comments through the rebuttal and subsequent responses, which addressed many of the concerns successfully with well structured discussions and additional experiments as well as results already reported in the appendix. Consequently, two reviewers championed this paper at the end of the post-rebuttal discussion. The other reviewer however kept their initial negative rating. The remaining concern of this reviewer centers on the marginal performance gain relative to the increased training complexity. While this is a valid comment, such a trade-off is inherently subjective; notably the other two reviewers valued the performance improvement achieved while imposing no test time computation. Hence, while acknowledging the reviewer's point, the AC believes this concern alone is not enough to justify rejecting the paper. Putting these together, the AC found that the positive points and the rebuttal outweigh the remaining concerns, and thus recommend acceptance of the paper. The authors are strongly encouraged to reflect the valuable comments from the reviewers and the additional results and discussions brought up during the rebuttal phase in the revision.